# Neural Dynamic Pricing: Provable and Practical Efficiency

## Abstract

Despite theoretical guarantees of existing dynamic pricing (DP) methods, their strong model assumptions may not reflect real-world conditions and are often unverifiable. This poses major challenges in practice since the performance of an algorithm may significantly degrade if the assumptions are not satisfied. Moreover, many DP algorithms show unfavorable empirical performance due to the lack of data efficiency. To address these challenges, we design a practical contextual DP algorithm that utilizes regression oracles. Our proposed algorithm assumes only Lipschitz continuity on the true conditional probability of purchase. We prove $\tilde{\mathcal{O}}(T^{\frac{2}{3}}\text{Regret}_R(T)^{\frac{1}{3}})$ regret upper bound where $T$ is the horizon and $\text{Regret}_R(T)$ is the regret of the oracle. The bound is nearly minimax optimal in the canonical case of finite function class, and our analysis generically applies to other function approximators including neural networks. To the best of our knowledge, our work is the first algorithm to utilize the powerful generalization capability of neural networks with provable guarantees in dynamic pricing literature. Extensive numerical experiments show that our algorithm outperforms existing state-of-the-art dynamic pricing algorithms in various settings, which demonstrates both provable efficiency and practicality.

## 1 Introduction

The contextual dynamic pricing (DP) is an online decision making and learning task where the seller sets prices for products based on contexts containing customer characteristics or product information. The seller tries to maximize the revenue by balancing between exploration and exploitation. In the widely studied customer valuation model setting, the binary choice of customer purchase $y_t$ is associated with market valuation $v_t$ to given context $x_t$; if the price $p_t$ set by the seller is cheaper than $v_t$, the seller receives $y_t = 1$ (purchase), otherwise $y_t = 0$ (non-purchase). The seller's goal is to maximize the revenue $\mathbb{E}[p_t y_t \,|\, x_t, p_t] = p_t \mathbb{P}(v_t > p_t)$. For the customer valuation $v_t$, the existing literature have considered various structures including linear (Amin et al., 2014; Javanmard & Nazerzadeh, 2019; Fan et al., 2022; Luo et al., 2022), log-linear (Shah et al., 2019), and proportional hazard model (Choi et al., 2023). When the noise distribution of the valuation, denoted by $F_0$, is assumed to be known (Javanmard & Nazerzadeh, 2019; Cohen et al., 2020; Xu & Wang, 2021), the algorithms proposed under such a setting are considered parametric. In contrast, semi-parametric algorithms (Shah et al., 2019; Fan et al., 2022; Luo et al., 2022; Xu & Wang, 2022; Choi et al., 2023) operate with unknown nonparametric $F_0$. The regret analysis of both parametric and semi-parametric methods often heavily rely on additional assumptions including log-concavity of $F_0$, smoothness of the revenue function, and stochastic conditions on contexts.

For this rigorous sequential decision-making problem, which also has significant practical impact, we pose the following fundamental question:

> **Question**: *Is there a practical contextual dynamic pricing method that works well across various instances while providing provable guarantees?*

To the best of our knowledge (and as many practitioners would agree), there has not been a single method that can confidently be said to perform robustly across various domains. As evident in the numerical experiments in Section 6, existing DP methods often fall short in terms of practical performance despite having provable guarantees. This contrasts sharply with the field of contextual

bandits. Although the two classes of problems appear similar, the practical effectiveness of contextual DP is still far from that of contextual bandits, where both provably efficient and practically effective methods are widely used (e.g., UCB (Li et al., 2010; Abbasi-Yadkori et al., 2011), TS (Chapelle & Li, 2011; Agrawal & Goyal, 2013), and IGW (Foster & Rakhlin, 2020)). Where do these discrepancies come from?

The reasons why the empirical performances of many of the DP methods with provable guarantees lack practical effectiveness are as follows: (i) Existing DP algorithms with provable guarantees do not utilize powerful function approximations such as neural networks, which may significantly improve practical performance. They are frequently restricted to strong assumptions about model or context distributions (Javanmard & Nazerzadeh, 2019; Fan et al., 2022). However, these assumptions may not be reflective of real-world conditions and, moreover, are often unverifiable in practical scenarios. The discrepancy between theoretical assumptions and practice can lead to significant performance degradation when these assumptions fail to hold true. This unreliable binary outcome—where algorithms perform well under ideal conditions but falter otherwise—poses a substantial challenge from a practitioner's perspective. (ii) Even when assumptions are satisfied, many of the existing DP methods still suffer from degradation in practical performance due to a lack of data efficiency. This is in fact the aspect that makes many contextual DP algorithm not as robust as contextual bandit algorithms. Many DP algorithms are based on epoch-wise estimation strategies (Javanmard & Nazerzadeh, 2019; Fan et al., 2022; Luo et al., 2022; Choi et al., 2023), where only at most half of the accumulated data can be utilized in any given epoch, throwing away at least half of the data. Moreover, they rely on exploration strategies such as epoch-wise explore-then-commit (Fan et al., 2022; Luo et al., 2022) or $\varepsilon$-greedy (Choi et al., 2023) (or some methods even never explore (Javanmard & Nazerzadeh, 2019)). This may lead to inefficient exploration since their epoch-wise exploration strategies are not fully adaptive to past history.

In this work, we propose `DP-IGW` (Dynamic Pricing with Inverse Gap Weighting), a practical dynamic pricing algorithm that leverages regression oracles including *neural regression* oracles, with provable guarantees. Our method applies to more general problem settings than existing approaches without sacrificing performance for theoretical rigor. We demonstrate that our algorithm significantly outperforms existing methods across a range of environments, including both simulation and real-world data, as demonstrated in Section 6. A key factor in its empirical success is the use of flexible regression oracles, such as neural networks, enabling general function approximation in diverse scenarios. Our analysis is based on the *generic binary choice model* instead of assuming customer valuation. In this model, the probability of purchase is a Lipschitz function of context and price: $y_t \mid x_t, p_t \sim \text{Ber}(f^\star(x_t, p_t))$ for some $f^\star(\cdot, \cdot)$. This *minimal assumption* of Lipschitz continuity captures complex real-world demand structures, with the customer valuation model as a special case. Our `DP-IGW` algorithm explores the price space using inverse gap weighting (IGW), a state-of-the-art exploration technique in contextual bandits due to its flexibility and minimal assumptions. (Abe & Long, 1999; Foster & Rakhlin, 2020; Foster & Krishnamurthy, 2021; Zhu & Mineiro, 2022). To the best of our knowledge, our work is the first adaptation of the IGW technique to dynamic pricing problems. The key difference between our algorithm and the IGW-based contextual bandit algorithms is that we separate the regression target (purchase) and exploration target (revenue), which facilitates efficient learning and exploration simultaneously in the DP setting. This efficiency, combined with the model flexibility, leads to remarkable empirical performance in diverse settings.

Our contributions are summarized as follows:

- We design `DP-IGW`, a contextual dynamic pricing algorithm operating under the generic binary choice model. Our algorithm outperforms the existing methods on almost all instances, often by significant margins— even in the instances where the assumptions of the existing methods are satisfied to their advantage. To our knowledge, our set of experiments in Section 6 present the most extensive and comprehensive evaluations so far in the contextual DP literature. Thus, our proposed algorithm achieves the best practicality among dynamic pricing algorithms with provable guarantees.

- Given a regression oracle with bounded regret $\text{Regret}_R(T)$, the algorithm guarantees $\tilde{\mathcal{O}}(T^{\frac{2}{3}} \text{Regret}_R(T)^{\frac{1}{3}})$ regret (Theorem 5.3), with a minimal assumption of Lipschitz continuity and adversarial contexts. Our proposed algorithm and its regret analysis are generic and applicable to any function approximator with guarantees. For example, we can utilize a neural network oracle, which makes our work to be the first provably efficient DP algo-

rithm utilizing neural networks, with $\tilde{\mathcal{O}}(T^{\frac{2}{3}})$ regret. The $\tilde{\mathcal{O}}(T^{\frac{2}{3}})$ rate is sharp compared to existing regret bounds, even under our relaxed assumptions. Neural networks provide high expressivity and generalization capability, as our numerical experiments demonstrate.

- For the canonical case of finite function classes, we prove a lower bound $\Omega(T^{\frac{2}{3}} \log^{\frac{1}{3}}(|\mathcal{F}|))$ (Theorem 5.5) that matches the upper bound up to logarithmic factors, establishing a nearly minimax optimal rate.

## 2 RELATED WORKS

**Contextual Dynamic Pricing with Binary Feedback.** A popular model in the literature is the linear valuation model where $\mathbb{P}(y = 1 \mid x, p) = 1 - F_0(p - \beta^T x)$. Some works assume known $F_0$ (Javanmard & Nazerzadeh, 2019; Xu & Wang, 2021; Cohen et al., 2020), estimating only the model parameter $\beta$. Javanmard & Nazerzadeh (2019) achieve $\mathcal{O}(s \log T)$ regret where $s$ is the number of nonzero elements in $\beta$, assuming that $F_0$ is log-concave, second-order differentiable, and stochastic contexts. Their algorithm design is epoch-based, which divides the entire time horizon into multiple clusters of consecutive steps ("epochs"), where the size of epoch is doubled. Within each epoch, the estimate of $\beta$ is fixed to a regularized maximum likelihood estimator using the data of the preceding epoch. Xu & Wang (2021) propose an online Newton step-based algorithm that operates with adversarial contexts, under strict log-concavity of $F_0$. Cohen et al. (2020) establish $\mathcal{O}(T^{\frac{2}{3}} d^{\frac{19}{6}})$ regret with a sub-Gaussian $F_0$ and adversarial contexts. Their algorithm searches for the optimal price by updating uncertainty sets of Löwner-John ellipsoids. On the other hand, some works assume different noise models, including zero noise (Amin et al., 2014; Leme & Schneider, 2018; Liu et al., 2021) where $y_t = \mathbb{I}_{\beta^T x_t > p_t}$, and constrained adversarial noise in Krishnamurthy et al. (2021).

Several semi-parametric DP algorithms assume unknown nonparametric $F_0$ on the linear valuation model (Fan et al., 2022; Luo et al., 2022; Xu & Wang, 2022). Fan et al. (2022) develop an epoch-based algorithm that estimates $\beta$ and $F_0$ each using the least-square and Nadaraya-Watson kernel regression (Nadaraya, 1964; Watson, 1964). It achieves $\tilde{\mathcal{O}}((Td)^{\frac{2m+1}{4m-1}})$ assuming that the optimal price is unique and the $F_0$ is $m \geq 2$ times differentiable. Luo et al. (2022) propose another epoch-based algorithm by learning $F_0$ by a finite-arm bandit problem. They prove $\mathcal{O}(T^{\frac{2}{3}})$ regret under 2nd-order smoothness and Lipschitz continuity of $F_0$. Xu & Wang (2022) discretize the price and parameter space then execute the EXP4 (Auer et al., 2002) algorithm, which has $\mathcal{O}(T^{\frac{3}{4}} + d^{\frac{1}{2}} T^{\frac{5}{8}})$ regret bound. Also, they prove $\Omega(T^{\frac{2}{3}})$ lower bound.

Other different valuation models have been proposed for the semi-parametric DP problem. Shah et al. (2019) assume the log-linear model where $\mathbb{P}(y = 1 \mid x, p) = 1 - F_0(p \exp(-\beta^T x))$. They propose an arm-elimination algorithm on the discretized grids of price-parameter space based on the confidence bound. They assume that the expected revenue function is smooth, which implies the uniqueness of the optimal price. Choi et al. (2023) assume a proportional hazard model where $\mathbb{P}(y = 1 \mid x, p) = (1 - F_0(p))^{\exp(\beta^T x)}$ and an epoch-based pricing algorithm with $\varepsilon$-greedy strategy for exploration. They achieve $\mathcal{O}(T^{\frac{2}{3}} d)$ regret assuming $F_0$ is continuously differentiable and an optimal price exists within the support of the price search domain.

There is a line of work that makes only a Lipschitz continuity assumption to the model. Mao et al. (2018) assume Lipschitz valuation and zero noise, where $y_t = \mathbb{I}_{f(x_t) > p_t}$ and $f$ is Lipschitz. They propose an algorithm based on confidence sets and prove a minimax optimal $\mathcal{O}(T^{\frac{d}{d+1}})$ regret. The setting in Chen & Gallego (2021) is the closest to ours, as they assume $y_t \mid x_t, p_t \sim \text{Ber}(f(x_t, p_t))$ where the functions $p \mapsto pf(x, p)$ and $x \mapsto pf(x, p)$ are Lipschitz. Unlike our setting, they additionally assume second-order smoothness and unique maxima of expected revenue function. They achieve $\mathcal{O}(T^{\frac{d+2}{d+4}})$ minimax regret bound by dynamically maintaining a partition of the context space while estimating the optimal price for each partition.

**Contextual Bandits.** As several works have noted (Kleinberg & Leighton, 2003; Luo et al., 2022; 2023), a DP problem can be naively framed by a bandit problem if we view $p_t$ as the action and $p_t y_t$ as the reward at step $t$. However, it is challenging to adapt the contextual bandit to the contextual DP problem in a natural way: (i) Reward $p_t y_t$ is modeled indirectly through the conditional distribution of feedback $y_t$ in DP problems, while the reward itself is modeled in bandit problems. (ii) Since the

action space of $p_t$ is continuous, finite-armed bandit algorithms need a discretization of the price space. Therefore, bandit algorithms may suffer from inefficient learning when merely applied to DP problems, as we demonstrate in Section 6. In the following part, we review recent works related to the contextual bandit problem. Lipschitz bandit algorithms (Kleinberg, 2004; Slivkins, 2011; Li et al., 2019) assume the Lipschitz continuity of the expected reward. Slivkins (2011) report the $\Omega(T^{\frac{d+2}{d+3}})$ lower bound. In Section 5, we derive a lower bound achieving the same order. The neural bandit family (Zhou et al., 2020; Zhang et al., 2020; Ban et al., 2022) adopts neural networks in learning and/or exploration. Zhou et al. (2020) and Zhang et al. (2020) establish $\tilde{\mathcal{O}}(\tilde{d}\sqrt{T})$ regret where $\tilde{d}$ is the effective dimension of the neural tangent kernel (NTK) matrix. If an oblivious adversary can select the context, then $\tilde{d}$ grows rapidly, leading to linear regret for them (Deb et al., 2024). Ban et al. (2022) prove $\tilde{\mathcal{O}}(\sqrt{T})$ regret which hides the dependence on the number of arms in logarithmic terms under stochastic i.i.d. contexts. Regression oracle-based contextual bandits (Foster & Rakhlin, 2020; Foster & Krishnamurthy, 2021; Zhu & Mineiro, 2022) assume that an off-the-shelf online supervised learning method is accessible for learning the reward model. Foster et al. (2020) assume that the oracle optimizes the squared loss and propose the inverse gap weighting exploration. Foster & Krishnamurthy (2021) employ an oracle for logarithmic loss to obtain first-order regrets under small losses, while Zhu & Mineiro (2022) propose a smoothed variant of the IGW technique to adapt to continuous action spaces. However, as mentioned, simply adapting these bandit algorithms to DP problems results in suboptimal performance due to the disregard for the unique structure of DP and the need for discretization.

## 3 PRELIMINARIES

**Problem Setting.** We formally specify our setting. The seller interacts with the customer for $T$ consecutive steps: (1) At step $t$, the seller observes a context $x_t \in \mathcal{X}$. (2) The seller sets a price $p_t \in \mathcal{P}$ based on $x_t$ and the previous history. (3) The seller observes the binary response $y_t|x_t, p_t \sim \mathrm{Ber}(f^\star(x_t, p_t))$ whether the product was sold or not. We assume that the contexts are determined by an oblivious adversary, and the price is chosen from a bounded interval $\mathcal{P} = [0, 1]$. Initially, the valuation function $f^\star$ is not known to the seller. The expected revenue of the price $p$ given the context $x_t$ is $\mathbb{E}[py_t|x_t, p] = pf^\star(x_t, p)$. Thus we define optimal price at time $t$ by $p_t^* \in \arg\max_p pf^\star(x_t, p)$, where there might exist multiple optimal prices. The goal of the seller is to maximize the expected revenue for $T$ steps, which is equivalent to minimizing the regret,

$$\mathrm{Regret}(T) := \mathbb{E}\left[\sum_{t=1}^{T}\{p_t^* f^\star(x_t, p_t^*) - p_t f^\star(x_t, p_t)\}\right]. \tag{1}$$

**Regression Oracle.** We assume that the seller can access a regression oracle $\mathrm{Alg}_R$ that sequentially estimates the probability of purchase. At each step $t$, $\mathrm{Alg}_R$ receives the previous history $\mathcal{H}_{t-1} := \{(x_i, p_i, y_i)\}_{i=1}^{t-1}$ as input and returns $\hat{f}_t$, an estimator of $f^\star$. We denote the fitted probability as $\hat{y}_t = \hat{f}_t(x_t, p_t)$. The goal of the oracle is to minimize the cumulative loss $\sum_{i=1}^{t}\ell_i(f)$ over $f \in \mathcal{F}$, where we consider logarithmic loss $\ell_i(f) = -y_i \log(f(x_i, p_i)) - (1 - y_i)\log(1 - f(x_i, p_i))$ or square loss $\ell_i(f) = (f(x_i, p_i) - y_i)^2$. We assume that $\mathrm{Alg}_R$ guarantees bounded regret:

**Assumption 3.1.** *(Online regression oracle) The regression oracle $\mathrm{Alg}_R$ guarantees that for any sequence $\mathcal{H}_t$, the regret of $\mathrm{Alg}_R$ is bounded by some data-independent function $\mathrm{Regret}_R(t)$,*

$$\mathbb{E}\left[\sum_{i=1}^{t}\ell_i(\hat{f}_i) - \inf_{f \in \mathcal{F}}\sum_{i=1}^{t}\ell_i(f)\right] \leq \mathrm{Regret}_R(t).$$

There is a large body of literature developing online regression algorithms for various function classes $\mathcal{F}$ such as finite function class (Vovk, 1995), generalized linear functions (Jézéquel et al., 2020), nonparametric regression (Gaillard & Gerchinovitz, 2015), and neural networks (Chen et al., 2021; Deb et al., 2024).

## 4 ALGORITHM

We describe the details of the proposed `DP-IGW` algorithm. Algorithm 1 states the pseudocode with online regression oracle.

---

**Algorithm 1** DP-IGW

---

1: **Input:** Regression oracle $\text{Alg}_R$, exploration parameter $\gamma_0$
2: **for** Epoch $k = 1, 2, \ldots$ **do**
3:     Set epoch length $\tau_k = 2^k$ and epoch index set $\mathcal{E}_k = \{\sum_{r=1}^{k-1} \tau_r + 1, \ldots, \sum_{r=1}^{k} \tau_r\}$
4:     Set exploration parameter $\gamma_k = \gamma_0 \cdot \tau_k^{1/3} \text{Regret}_R (2\tau_k - 2)^{-1/3}$
5:     **for** $t \in \mathcal{E}_k$ **do**
6:         Observe $x_t$ and access $\hat{f}_t(x_t, \cdot)$ via $\text{Alg}_R$
7:         Compute $\hat{p}_t \leftarrow \arg\max_{p \in \mathcal{P}} p\hat{f}_t(x_t, p)$, then sample $p_t \sim P_t = M_t + (1 - M_t(\mathcal{P}))\mathbb{I}_{\hat{p}_t}$
8:         Set price $p_t$ and observe $y_t$, then update $\text{Alg}_R$ with $(x_t, p_t, y_t)$
9:     **end for**
10: **end for**

---

**Exploration via Inverse Gap Weighting.**   At each step, the online regression oracle predicts the probability of purchase $\hat{f}(x_t, \cdot)$. The greedy price is computed as $\hat{p}_t \leftarrow \arg\max_{p \in \mathcal{P}} p\hat{f}_t(x_t, p)$ based on the prediction, then the price $p_t$ is sampled from the distribution $P_t = M_t + (1 - M_t(\mathcal{P}))\mathbb{I}_{\hat{p}_t}$

$$\text{where} \quad M_t(\omega) = \int_{p \in \omega} m_t(p) d\mu(p), \quad m_t(p) = \frac{1}{1 + \gamma_k(\hat{p}_t \hat{f}_t(x_t, \hat{p}_t) - p\hat{f}_t(x_t, p))}. \tag{2}$$

The hyperparameter $\gamma_t > 0$ determines the degree of exploration, which is discussed in the next paragraph. Given a sampling oracle for drawing samples from $\mu$, the sampling from $P_t$ can be efficiently done by rejection sampling. The construction of $P_t$ is based on the SmoothIGW algorithm in Zhu & Mineiro (2022), which extends the inverse gap weighting (IGW) technique (Abe & Long, 1999; Foster & Rakhlin, 2020) to continuous spaces. Specifically, $P_t$ is a mixture of the atomic distribution $\mathbb{I}_{\hat{p}_t}$ and the distribution induced by the density $m_t(p)$ and the base measure $\mu$. Note that $m_t(p) \leq 1$ for all $p \in \mathcal{P}$, thus $M_t(\mathcal{P}) \leq 1$ and $P_t$ is well-defined. The density $m_t(p)$ assigns probability inversely proportional to the gap in the estimated revenue, between the greedy price $\hat{p}_t$ and price $p$. Intuitively, the IGW smoothly balances exploration and exploitation by placing more weight on the region of high estimated revenue.

**Scheduling the Degree of Exploration.**   We divide the horizon $T$ into several epochs, where $k$-th epoch spans $2^k$ steps. In each epoch, we set the exploration hyperparameter $\gamma_k$ to construct the sampling distribution equation 2. Our proposed scheduling of $\gamma_k$ enables DP-IGW to be an anytime algorithm. If we fix the exploration parameter as $\gamma_k = \gamma$ for $T$ steps, $\gamma$ must depend on the horizon $T$ to ensure bounded regret. However, with the proposed scheduling, $\gamma_k$ depends only on the current epoch length $\tau_k$, thus the algorithm is executable without knowing the horizon beforehand.

**Remark.**   The DP-IGW algorithm is *fully sequential*; we do not discard any past data, and *fully adaptive*; we explore adaptively based on full data, as opposed to many existing epoch-based dynamic pricing algorithms (Javanmard & Nazerzadeh, 2019; Fan et al., 2022; Luo et al., 2022; Choi et al., 2023). Unlike the standard doubling trick (Cesa-Bianchi & Lugosi, 2006), our use of epoch is only for scheduling the value of $\gamma_k$. The computational complexity of DP-IGW is determined by the argmax operation (Line 6) and oracle update (Line 8), as the sampling of $p_t$ can be done by rejection sampling at a constant cost per step. The argmax operation and the oracle update depend on function class $\mathcal{F}$, and we note that almost every DP algorithm has the argmax operation for finding the greedy price, thus DP-IGW is no worse than existing DP algorithms with respect to the argmax operation. The oracle update also maintains feasible complexity if we use neural regression algorithms (Deb et al., 2024) which is based on standard gradient descent. Therefore, our algorithm is computationally efficient.

## 5 THEORETICAL ANALYSIS

We derive a regret upper bound of DP-IGW with an online regression oracle satisfying Assumption 3.1. Furthermore, we prove the regret lower bound of the generic binary choice model under the Lipschitz continuity Assumption 5.2.

## 5.1 REGRET UPPER BOUND

**Assumption 5.1.** *The true valuation function $f^\star$ is realizable, i.e. $f^\star \in \mathcal{F}$.*

**Assumption 5.2.** *There exists some constant $L > 0$ such that, for any $x \in \mathcal{X}$ and any $p_1, p_2 \in \mathcal{P}$, it holds that $|f^\star(x, p_1) - f^\star(x, p_2)| \le L|p_1 - p_2|$.*

We make two assumptions to derive the regret upper bound. Assumption 5.1 is a common one in dynamic pricing and bandit literature, indicating that we are working within the realizable setting. Assumption 5.4 is a Lipschitz continuity on the purchase with respect to the price domain, which is a considerably weaker assumption compared to those in previous studies under the customer valuation models. Now we present our main theorem, whose detailed proof is deferred to Appendix A.

**Theorem 5.3.** *Under Assumption 3.1, 5.1, and 5.2, setting $\gamma_0 = \Theta((L+1)^{-\frac{1}{3}})$, Algorithm 1 guarantees*

$$\text{Regret}(T) \le \tilde{\mathcal{O}}\left(T^{\frac{2}{3}} \cdot \text{Regret}_R(T)^{\frac{1}{3}}\right).$$

**Discussion on Theorem 5.3.** Consider a finite function class $\mathcal{F}$. Vovk's aggregation algorithm (Vovk, 1995) ensures $\text{Regret}_R(T) \le \log|\mathcal{F}|$. Plugging this into Theorem 5.3, DP-IGW guarantees $\tilde{\mathcal{O}}(T^{\frac{2}{3}}\log^{\frac{1}{3}}(|\mathcal{F}|))$ regret bound. This matches the lower bound of Theorem 5.5 up to logarithmic factors, thus we obtain nearly minimax optimal regret guarantee. We emphasize that DP-IGW guarantees regret upper bound for any $\mathcal{F}$ satisfying Assumption 5.1 and 5.4 if we have a regression oracle for $\mathcal{F}$. The neural networks are arguably the most practical instance of $\mathcal{F}$ for which we have a regression oracle. The online regression method in Deb et al. (2024) has a regret bound of $\text{Regret}_R(t) \le \mathcal{O}(\log t)$ for a certain class of neural networks. Using the neural regression algorithm in Deb et al. (2024), DP-IGW achieves $\tilde{\mathcal{O}}(T^{\frac{2}{3}})$ regret. Remarkably, the $\tilde{\mathcal{O}}(T^{\frac{2}{3}})$ rate is sharper than or matches the regret bound of semi-parametric DP algorithms (Shah et al., 2019; Fan et al., 2022; Luo et al., 2022; Xu & Wang, 2022; Choi et al., 2023), despite DP-IGW is based on the more expressive model and the weaker assumption: $\mathcal{F}$ is the set of functions realizable by neural networks whose input is $(x_t, p_t)$, and the contexts are adversarial. The detail on the neural regression oracle is explained in Appendix D. There are other efficient regression oracles on different function classes, including logistic regression (Jézéquel et al., 2020), nonparametric regression (Gaillard & Gerchinovitz, 2015), and kernel regression (Jézéquel et al., 2019), to list a few.

**Remark.** Our result in Theorem 5.3 holds for adversarial contexts, which is why the online regression oracle satisfying Assumption 3.1 is required. We note that Assumption 3.1 can be relaxed to an offline regression oracle in Assumption B.1 when the contexts are stochastic. In the offline setting, with probability $1 - \delta$, the oracle achieves an upper bound of learning guarantee $\mathcal{E}_{\mathcal{F},\delta}(n)$ which decreases with training dataset size $n$, hence we can derive a regret upper bound $\mathcal{O}(T \cdot \mathcal{E}_{\mathcal{F},\delta}(T)^{1/3})$ (Theorem B.2). The offline oracle version is computationally efficient in that the oracle update is only made at the end of each epoch, so the number of oracle updates is $\mathcal{O}(\log(T))$. The extension is based on the technique from Simchi-Levi & Xu (2022), and we discuss the offline version of DP-IGW in Appendix B. Consider a finite function class $\mathcal{F}$. Using the ERM(Empirical Risk Minimization) predictor of Theorem 7.6 in Van Erven et al. (2015) which guarantees $\mathcal{E}_{\mathcal{F},\delta}(n) = 2\log(|\mathcal{F}/\delta|)/n$, DP-IGW achieves $\tilde{\mathcal{O}}(T^{\frac{2}{3}})$ regret, which is a sharp rate as discussed above. We also consider $\mathcal{F}$ of neural networks. Assume $d$ be the context dimension and $\beta$ be the Sobolev ball smoothness containing $f^*$. Using the neural network estimator in Farrell et al. (2021) which guarantees $\mathcal{E}_{\mathcal{F},\delta}(n) = \tilde{\mathcal{O}}(n^{-\frac{\beta}{\beta+d}})$, our DP-IGW achieves $\tilde{\mathcal{O}}(T^{\frac{2\beta+3d}{3(\beta+d)}})$.

## 5.2 LOWER BOUND OF LIPSCHITZ DYNAMIC PRICING

We now present a lower bound for any dynamic pricing problem that assumes a Lipschitz continuity on the conditional probability of purchase, as below.

**Assumption 5.4.** *There exists some constant $L > 0$ such that, for any $x_1, x_2 \in \mathcal{X}$ and any $p_1, p_2 \in \mathcal{P}$, it holds that $|f^\star(x_1, p_1) - f^\star(x_2, p_2)| \le L(\|x_1 - x_2\|_2 + |p_1 - p_2|)$. In addition, for any $x \in \mathcal{X}$, it holds that $f^\star(x, p_1) \ge f^\star(x, p_2)$ whenever $p_1 < p_2$.*

Note that Assumption 5.4 is stronger than Assumption 5.2, where the former assumes Lipshitz continuity in both context and price space, but the latter assumes context-wise Lipschitz continuity. The monotonicity in Assumption 5.4 addresses a canonical axiom that demand would be monotone decreasing with respect to price. Theorem 5.5 establishes the lower bounds for this setting. The proof is based on the standard "needle in the haystack" instance (Auer et al., 2002; Kleinberg, 2004; Slivkins, 2011) and its extension to DP (Luo et al., 2022; Xu & Wang, 2022).

**Theorem 5.5.** *Under Assumption 5.4, for any dynamic pricing algorithm, there exists a problem instance that has regret* $\mathrm{Regret}(T) \geq \Omega(T^{\frac{d+2}{d+3}})$. *If the reference function class is finite, we have* $\mathrm{Regret}(T) \geq \Omega(T^{\frac{2}{3}} \log^{\frac{1}{3}}(|\mathcal{F}|))$.

**Discussion on Theorem 5.5.** Chen & Gallego (2021) establish $\tilde{\mathcal{O}}(T^{\frac{d+2}{d+4}})$ regret upper bound for Lipschitz dynamic pricing, with additional smoothness and local concavity assumptions. Our result shows that the assumptions they made have an impact on the complexity of learning. The $\Omega(T^{\frac{d+2}{d+3}})$ and $\Omega(T^{\frac{2}{3}} \log^{\frac{1}{3}}(|\mathcal{F}|))$ lower bound match the lower bound of Lipschitz contextual bandits (Slivkins, 2011; Krishnamurthy et al., 2020). This implies that contextual dynamic pricing is as hard as contextual bandits with the minimal assumption of Lipschitz continuity. Compared to the semi-parametric customer valuation models (Fan et al., 2022; Luo et al., 2022; Xu & Wang, 2022; Choi et al., 2023), their $\Omega(T^{\frac{2}{3}})$ lower bound implies that the semi-parametric DP is considerably simpler than the general Lipschitz DP. It is worth noting that the proof of the lower bound in Luo et al. (2022); Xu & Wang (2022) construct a non-contextual function class, albeit the model contains contexts. Therefore, their lower bound falls in the special case of our result, where $d = 0$. Finally, it is noteworthy that Theorem 5.3 does not require the monotonicity assumption, which implies that the monotonicity has no impact on the asymptotic complexity.

## 6 NUMERICAL EXPERIMENTS

We evaluate our `DP-IGW` algorithm via extensive numerical experiments including simulation environments and real data. We train a neural network oracle by minimizing logarithmic loss, as described in Appendix E. We compare our method to existing DP algorithms and contextual bandit algorithms (with proper modification), using cumulative regret as the performance metric.

**Baseline Methods: Dynamic Pricing** We first consider dynamic pricing methods with flexible model assumptions as baselines, as algorithms with strong assumptions are impractical due to the potential for model misspecification. Therefore, we select 5 semi-parametric or nonparametric dynamic pricing methods as baseline methods: ExUCB (Luo et al., 2022), Fan et al. (2022), DEEP-C (Shah et al., 2019), CoxCP (Choi et al., 2023), and ABE (Chen & Gallego, 2021). ExUCB and Fan et al. (2022) assume the linear valuation model, DEEP-C works on the log-linear model, and CoxCP is based on the PH model. ABE is a nonparametric DP algorithm. We then consider dynamic pricing methods which has prior information on the model with strong assumptions as baselines to verify whether our algorithm performs well despite such prior information. ONSP (Xu & Wang, 2021), RMLP (Javanmard & Nazerzadeh, 2019), and RMLP2 (Javanmard & Nazerzadeh, 2019) are parametric methods which assume the linear valuation model with known noise distribution $F_0$ and log-concavity of both $F_0$ and $1 - F_0$. We optimized hyperparameters for each method, see Appendix E for the details.

**Baseline Methods: Neural Bandits** Although contextual bandits are not equivalent to dynamic pricing problems in general, we can interpret dynamic pricing problems as contextual bandits with stochastic rewards. Considering the price $p_t$ as an action and the realized revenue $p_t y_t$ as a reward, the distribution of the reward is determined by the context $x_t$ and the action $p_t$. From this viewpoint, we can interpret the regret 1 as the pseudo-regret of the bandit problem. Therefore, we compare `DP-IGW` with recent state-of-the-art contextual bandit algorithms leveraging neural networks: NeuralUCB (Zhou et al., 2020), Neural Thompson sampling (NeuralTS), Zhang et al. (2020)), SquareCB (Foster & Rakhlin, 2020), and SmoothIGW (Zhu & Mineiro, 2022). Since NeuralUCB, NeuralTS, and SquareCB are finite-armed bandit algorithms, we evenly discretize the price with finite arms. The neural network structures for all algorithms are the same, as described in Appendix E.

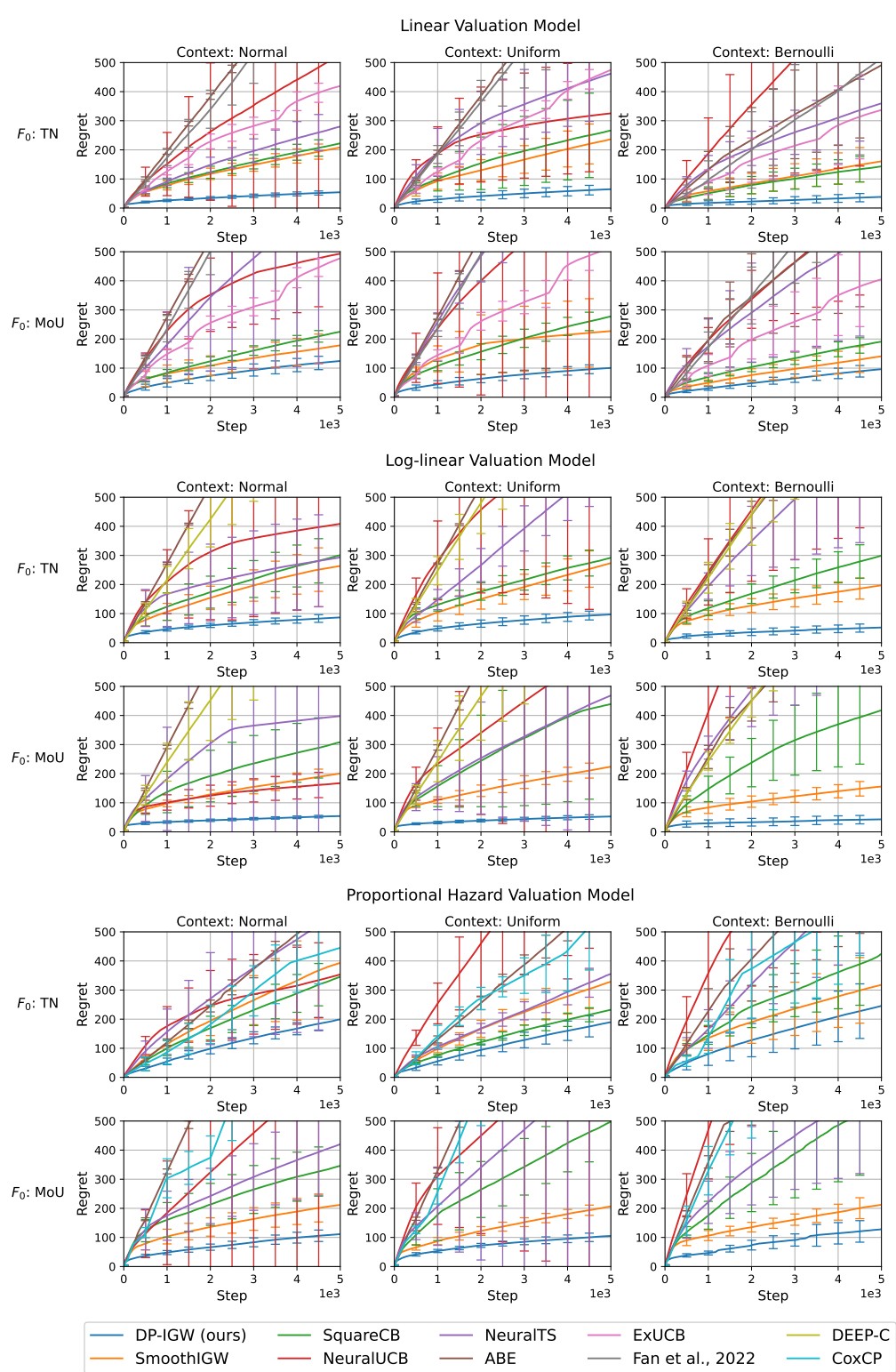

Figure 1: Cumulative regret of the algorithms in simulation environments, averaged over 5 experiments. Abbreviations each indicate TN: Truncated Normal, MoU: Mixture of Uniform.

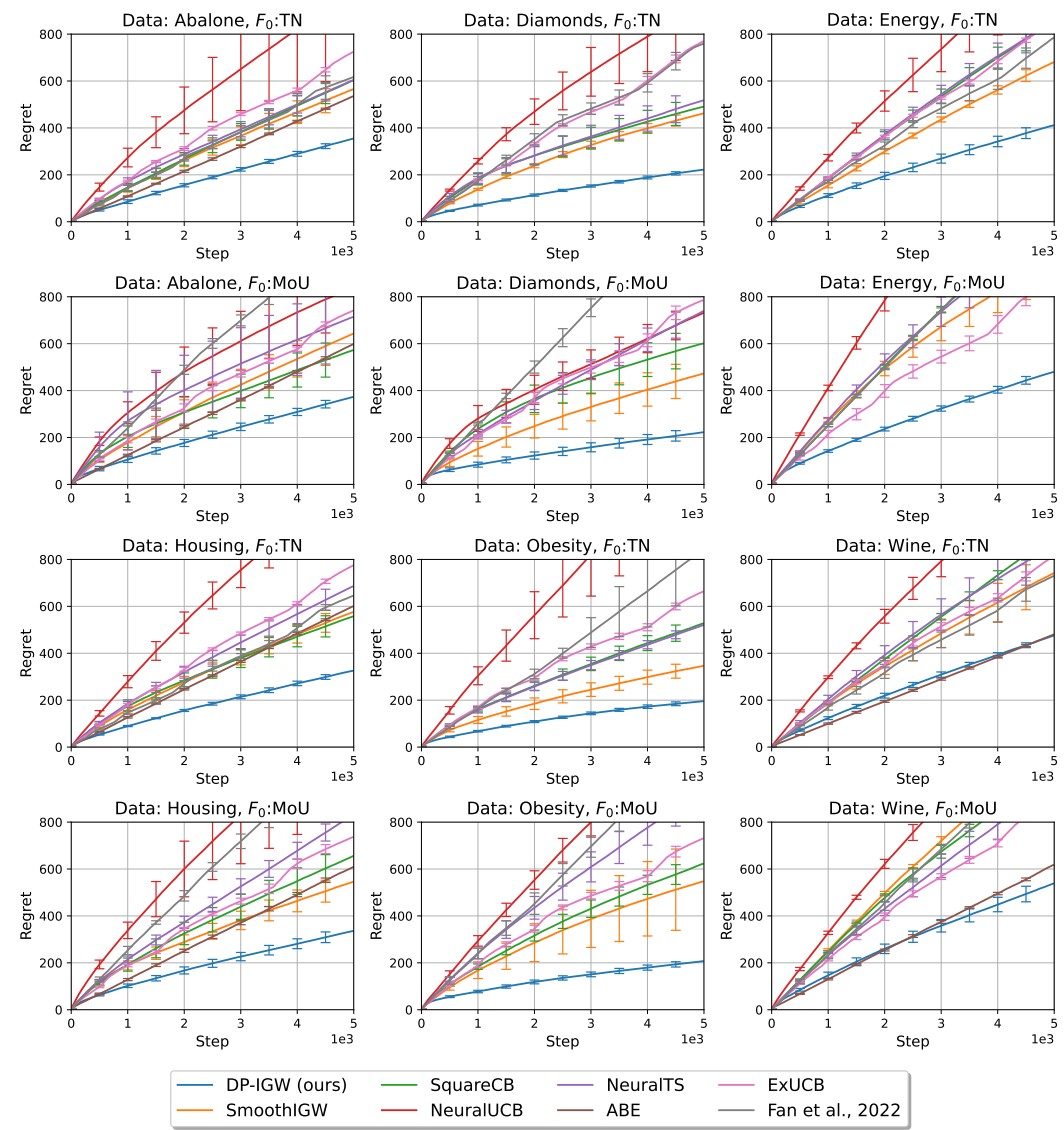

Figure 2: Cumulative regret of the algorithms with real data, averaged over 5 experiments. Abbreviations each indicate TN: Truncated Normal, MoU: Mixture of Uniform.

## 6.1 SIMULATION ENVIORNMENTS

We employ customer valuation models to configure simulation environments. Since semi-parametric DP methods assume different valuation models, we consider three valuation models $\mathbb{P}(v_t > p \mid x_t)$: linear valuation $1 - F_0(p - \beta^T x_t)$, log-linear valuation $1 - F_0(p \exp(-\beta^T x_t))$, and proportional hazard (PH) model $(1 - F_0(p))^{\exp(\beta^T x_t)}$. For each valuation model, we consider two base CDFs $F_0$: Truncated Normal and Mixture of Uniform. Contexts $x_t$ are sampled i.i.d. from three distributions: normal distribution, uniform in a unit ball, and Bernoulli distribution for all $i \in [d]$. In total, there are 6 combinations of $F_0$, context distributions pair for each valuation model. The model parameter is randomly sampled by $\beta \sim \mathcal{N}(0, \frac{1}{\sqrt{d}^2}I)$, with $d = 5$. Details on settings and hyperparameter search are provided in Appendix E.

As illustrated in Figure 1, `DP-IGW` significantly outperforms these algorithms in all settings. We emphasize that the parametric and semi-parametric dynamic pricing methods exploit the model

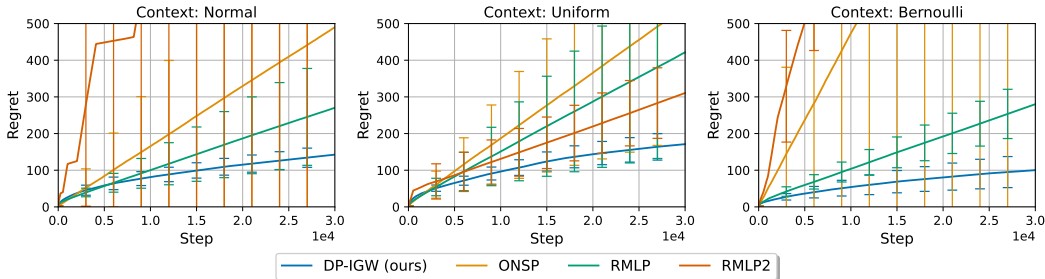

Figure 3: Cumulative regret (for $T = 30000$ steps) of parametric DP algorithms in simulation environments, averaged over 10 experiments. We experiment with the linear valuation model and normal CDF to satisfy the model assumptions.

structure, yet `DP-IGW` learns the valuation models better. Even if `DP-IGW` has no prior information on the model, it outperforms the parametric algorithms with a lower variance of performance as in Figure 3. We attribute the superior performance of `DP-IGW` to two factors: (i) Enhanced data-efficiency. Doubling-epoch-based semi-parametric algorithms discard past data at the initiation of epochs, utilizing less than half of the observed data up to the current step. In contrast, `DP-IGW` utilizes all data for learning. (ii) `DP-IGW` benefits from the generalization capacity of the neural network-based regression oracle, ensuring consistent performance across diverse environments. Moreover, `DP-IGW` smoothly balances exploration and exploitation, unlike epoch-based algorithms that show discontinuous transitions in the regret curves.

While the neural network is a factor in performance improvement, it does not explain everything, as `DP-IGW` consistently outperforms neural bandit algorithms. This results from the structure of dynamic pricing problems, where the reward (revenue) in the regret and the feedback (purchase) are defined in distinct ways. `DP-IGW` successfully exploits this structure by separating the learning target and the exploration target: $\mathrm{Alg}_R$ aims to estimate $f^\star(x_t, p_t)$, while $p_t$ is sampled inversely proportional to $p\hat{f}(x_t, p)$.

## 6.2 REAL-WORLD DATA

In the dynamic pricing literature, experiments with real-world data have been limited since complex underlying structures of real-world data may violate model assumptions. However, `DP-IGW` empowered by neural networks can successfully learn with real-world data, as we demonstrate in this section. We experiment with six real-world datasets for regression tasks: Abalone (Nash & Ford, 1994), Diamonds (Wickham, 2016), Appliance Energy Prediction (Energy) (Candanedo, 2017), Estimation of Obesity Levels (Obesity) (Palechor & De la Hoz Manotas, 2019), California Housing (Housing) (Pace & Barry, 1997), Wine Quality (Wine) (Cortez & Reis, 2009). The datasets contain continuous and categorical features, with dimensions ranging from 10 to 26 after one-hot encoding of the categorical features. Therefore, we can investigate the performances of algorithms in various real-world scenarios. Refer to Appendix E.1 for the details on the datasets. We left out DEEP-C because its computational cost increases exponentially with $d$, and also excluded CoxCP as it can't estimate model parameters when dealing with categorical features.

To simulate the online interaction of dynamic pricing problems, we treat the regression targets of the datasets as valuations. At step $t$, one context vector $x_t$ and corresponding valuation (regression target) $v_t$ is sampled from the dataset. The algorithm sets the price $p_t$ based on $x_t$, then $y_t$ is sampled from $\mathrm{Ber}(1 - F_0(p_t - v_t))$. As in the simulation experiments, we consider two options for $F_0$.

Figure 2 shows that `DP-IGW` has the best performance among the baselines, in almost every dataset. This shows that `DP-IGW` can efficiently learn complex real-world data. Also, it scales well with the dimension of contexts and the size of datasets.

## 7 REPRODUCIBILITY

We provide detailed descriptions of the experiments, including training protocol and neural network architecture, in Section 6 and Section E. Supplementary materials include the code used to run the experiments, instructions for setting up the environment, commands to run experiments, and code for generating the figures. Additionally, the processed real-world dataset and the code for data processing is included in the supplementary material.

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

# A PROOF OF THEOREM 5.3

In this section, we present the detailed proof of Theorem 5.3. Our proof is based on the IGW technique in Zhu & Mineiro (2022), and the following definition of smooth regret plays an important role:

**Definition A.1.** *Let $(\Omega, \mathcal{G}, \mu)$ be a probability space. For $h > 0$, define the set of $h$-smooth kernels $\mathcal{Q}_h$ such that any $Q \in \mathcal{Q}_h$ satisfies: (i) Absolute continuity $Q \ll \mu$ (ii) Bounded Radon-Nikodym derivative $\frac{dQ}{d\mu} \leq 1/h$. For any $x \in \mathcal{X}$, the largest revenue that can be achieved by $h$-smooth kernel is $\text{Smooth}_h(x) := \sup_{Q \in \mathcal{Q}_h} \mathbb{E}_{p \sim Q}[pf^\star(x, p)]$. Based on that, the smooth regret is defined as: $\text{Regret}_h(T) := \mathbb{E}[\sum_{t=1}^T \text{Smooth}_h(x_t) - p_t f^\star(x_t, p_t)]$.*

The smooth regret $\text{Regret}_h(T)$ is a "smoothed" version of $\text{Regret}(T)$, in that the algorithm competes with the best $h$-smooth policy instead of the best atomic policy $\mathbb{I}_{p_t^*}$. Under a Lipschitz continuity assumption on $f^\star$, a bound on the smooth regret can be transformed into a bound on the standard regret by Lemma A.2. Overall, the smooth regret is a stepping stone for the standard regret bound.

**Lemma A.2.** *For any $h \in (0, 1]$ and any episode $\mathcal{E}_k$, we have*

$$\text{Regret}(\mathcal{E}_k) \leq \text{Regret}_h(\mathcal{E}_k) + (L+1)h\tau_k. \tag{3}$$

*where*

$$\text{Regret}(\mathcal{E}_k) := \sum_{t \in \mathcal{E}_k} \mathbb{E}\left[p_t^* f^\star(x_t, p_t^*) - p_t f^\star(x_t, p_t)\right]$$

*is the standard regret within $\mathcal{E}_k$ and*

$$\text{Regret}_h(\mathcal{E}_k) := \sum_{t \in \mathcal{E}_k} \mathbb{E}\left[\text{Smooth}_h(x_t) - p_t f^\star(x_t, p_t)\right]$$

*is the smooth regret within $\mathcal{E}_k$.*

*Proof.* We employ the discretization argument of Zhu & Mineiro (2022). Fix any $f \in \mathcal{F}$, and divide the price set $\mathcal{P} = [0, 1]$ into $B = \lceil \frac{1}{h} \rceil$ intervals $\{I_b\}_{b=1}^B$ such that $I_b = [(b-1)h, bh)$, and $b^*$ is the interval containing the optimal price $p^*$, i.e. $p^* = \arg\max_{p \in \mathcal{P}} pf(x_t, p) \in I_{b^*}$. Observe that

$$\begin{aligned} |p^* f(x_t, p^*) - pf(x_t, p)| &= |p^*(f(x_t, p^*) - f(x_t, p)) + (p^* - p)f(x_t, p)| \\ &\leq |p^*| |f(x_t, p^*) - f(x_t, p)| + |p^* - p| |f(x_t, p)| \leq (L+1) |p^* - p| \end{aligned} \tag{4}$$

due to triangular inequality, $|p^*| \leq 1$, $|f(x_t, p)| \leq 1$, and Assumption 5.4. Setting the smoothing kernel $\text{Unif}(I_{b_t}) \in \mathcal{Q}_h$ as a reference policy, we have

$$p^* f(x_t, p^*) \leq \mathbb{E}_{p \sim \text{Unif}(I_{b_t})} [pf(x_t, p)] + (L+1)h \tag{5}$$

$$\leq \sup_{Q \in \mathcal{Q}_h} \mathbb{E}_{p \sim Q} [pf(x_t, p)] + (L+1)h = \text{Smooth}_h(x_t) + (L+1)h. \tag{6}$$

Since equation 5 holds for $f^\star$, taking summation over $t \in \mathcal{E}_k$, we have

$$\text{Regret}(\mathcal{E}_k) \leq \text{Regret}_h(\mathcal{E}_k) + (L+1)h\tau_k. \tag{7}$$

$\square$

Now we introduce the Decision-Estimation Coefficient (DEC, Foster & Rakhlin (2020); Zhu & Mineiro (2022); Foster & Krishnamurthy (2021)):

**Definition A.3.** *For $\gamma > 0$ and a function class $\mathcal{F}$, define $\text{dec}_\gamma(\mathcal{F}) := \sup_{\hat{f}, x} \text{dec}_\gamma(\mathcal{F}, \hat{f}, x)$ where*

$$\text{dec}_\gamma(\mathcal{F}, \hat{f}, x) :=$$

$$\inf_{P \in \Delta(\mathcal{P})} \sup_{f^\star \in \mathcal{F}} \mathbb{E}_{p \sim P} \left[\text{Smooth}_h(x) - pf^\star(x, p) - \frac{\gamma}{4h} \left(f^\star(x, p) - \hat{f}(x, p)\right)^2\right].$$

By the definition of $\text{Smooth}_h(x) = \sup_{Q \in Q_h} \mathbb{E}_{p \sim Q}[pf^\star(x, p)]$, it is easy to see that $\text{dec}_\gamma(\mathcal{F}, \hat{f}, x)$ can be expressed as another form:

$$\inf_{P \in \Delta(\mathcal{P})} \sup_{Q \in Q_h} \sup_{f^\star \in \mathcal{F}} \mathbb{E}_{p \sim P, p^\star \sim Q} \left[ p^\star f^\star(x, p^\star) - pf^\star(x, p) - \frac{\gamma}{4h} \left( f^\star(x_t, p_t) - \hat{f}(x_t, p_t) \right)^2 \right].$$

The following two lemmas from Zhu & Mineiro (2022) guarantee that $\text{dec}_\gamma(\mathcal{F})$ is upper bounded. We include the proof for completeness.

**Lemma A.4.** *For any $\gamma > 0$ and $h \in (0, 1]$, it holds that*

$$dec_\gamma(\mathcal{F}) \leq \frac{3}{\gamma}$$

*under sampling distribution $P = M + (1 - M(\mathcal{P})) \cdot \mathbb{I}\{\hat{p}\}$ with*

$$\hat{p} = \arg\max_{p \in \mathcal{P}} p\hat{f}(x, p), \quad M(\omega) = \int_{p \in \omega} m(p) d\mu(p), \quad m(p) = \frac{1}{1 + \gamma(\hat{p}\hat{f}(x, \hat{p}) - p\hat{f}(x, p))}$$

*Proof.* Fix $Q \in \mathcal{Q}_h$ and $f \in \mathcal{F}$, then simplify the notations as $f(p) := f(x, p)$ and $\hat{f}(p) := \hat{f}(x, p)$. It holds that

$$\mathbb{E}_{p \sim P, p^\star \sim Q} \left[ p^\star f(p^\star) - pf(p) - \frac{\gamma}{4h}(f(p) - \hat{f}(p))^2 \right]$$

$$\leq (1 - M(\mathcal{P})) \left( -\hat{p}f(\hat{p}) - \frac{\gamma}{4h}(f(\hat{p}) - \hat{f}(\hat{p}))^2 \right) \tag{8}$$

$$+ \mathbb{E}_{p \sim M, p^\star \sim Q} \left[ p^\star f(p^\star) - pf(p) - \frac{\gamma}{4h}(f(p) - \hat{f}(p))^2 \right]$$

by the definition of $P$. For the first term, it holds that

$$(1 - M(\mathcal{P})) \left( -\hat{p}f(\hat{p}) - \frac{\gamma}{4h}(f(\hat{p}) - \hat{f}(\hat{p}))^2 \right)$$

$$= (1 - M(\mathcal{P})) \left( -\hat{p}\hat{f}(\hat{p}) - \hat{p}(f(\hat{p}) - \hat{f}(\hat{p})) - \frac{\gamma}{4h}(f(\hat{p}) - \hat{f}(\hat{p}))^2 \right) \tag{9}$$

$$\leq (1 - M(\mathcal{P})) \left( -\hat{p}\hat{f}(\hat{p}) + \frac{h\hat{p}^2}{\gamma} \right).$$

By the definition of $\mathcal{Q}_h$ and the fact that $m(p) > 0$, we have $Q \ll \mu$ and $\mu \ll M$, hence $Q \ll M$. Therefore, Lemma A.5 implies that

$$\mathbb{E}_{p \sim M, p^\star \sim Q} \left[ p^\star f(p^\star) - pf(p) - \frac{\gamma}{4h}(f(p) - \hat{f}(p))^2 \right]$$

$$\leq \mathbb{E}_Q \left[ p\hat{f}(p) \right] - \mathbb{E}_M \left[ p\hat{f}(p) \right] + \frac{h}{\gamma} \mathbb{E}_M \left[ \left( \frac{q(p)}{m(p)} - 1 \right)^2 \right]$$

$$\leq \mathbb{E}_Q \left[ p\hat{f}(p) \right] - \mathbb{E}_M \left[ p\hat{f}(p) \right] + \frac{h}{\gamma} \mathbb{E}_\mu \left[ \frac{q^2(p)}{m(p)} - 2q(p) + m(p) \right] \tag{10}$$

$$\leq \mathbb{E}_Q \left[ p\hat{f}(p) \right] - \mathbb{E}_M \left[ p\hat{f}(p) \right] + \frac{h}{\gamma} \mathbb{E}_\mu \left[ \frac{q^2(p)}{m(p)} \right] + \frac{hM(\mathcal{P})}{\gamma}$$

$$= -\mathbb{E}_M \left[ p\hat{f}(p) \right] + \frac{1}{\gamma} \mathbb{E}_\mu \left[ q(p) \left( \gamma p\hat{f}(p) + h\frac{q(p)}{m(p)} \right) \right] + \frac{hM(\mathcal{P})}{\gamma}$$

Now we bound each term in equation 10. First, by definition of $M$, the first term of equation 10 is bounded by:

$$-\mathbb{E}_M \left[ p\hat{f}(p) \right] = -\mathbb{E}_\mu \left[ \frac{p\hat{f}(p)}{1 + \gamma(\hat{p}\hat{f}(\hat{p}) - p\hat{f}(p))} \right]$$

$$= \frac{1}{\gamma} \mathbb{E}_\mu \left[ -\frac{\gamma\hat{p}\hat{f}(\hat{p})}{1 + \gamma(\hat{p}\hat{f}(\hat{p}) - p\hat{f}(p))} + \frac{\gamma(\hat{p}\hat{f}(\hat{p}) - p\hat{f}(p))}{1 + \gamma(\hat{p}\hat{f}(\hat{p}) - p\hat{f}(p))} \right] \tag{11}$$

$$= \frac{1}{\gamma} \mathbb{E}_\mu \left[ -\gamma\hat{p}\hat{f}(\hat{p})m(p) + (1 - m(p)) \right] = -\hat{p}\hat{f}(\hat{p})M(\mathcal{P}) + \frac{1 - M(\mathcal{P})}{\gamma}$$

The second term of equation 10 is bounded by:

$$\frac{1}{\gamma}\mathbb{E}_\mu\left[q(p)\left(\gamma p\hat{f}(p) + h\frac{q(p)}{m(p)}\right)\right] = \frac{1}{\gamma}\mathbb{E}_\mu\left[q(p)\left(\gamma p\hat{f}(p) + hq(p)\left(1 + \gamma(\hat{p}\hat{f}(\hat{p}) - p\hat{f}(p))\right)\right)\right]$$

$$\leq \frac{1}{\gamma}\mathbb{E}_\mu\left[q(p)\left(\gamma p\hat{f}(p) + 1 + \gamma(\hat{p}\hat{f}(\hat{p}) - p\hat{f}(p))\right)\right] = \frac{1}{\gamma}\mathbb{E}_\mu\left[q(p)\left(1 + \gamma\hat{p}\hat{f}(\hat{p})\right)\right] \leq \frac{1}{\gamma} + \hat{p}\hat{f}(\hat{p})$$

$$(12)$$

where the first inequality holds due to the fact that $\hat{p}\hat{f}(\hat{p}) - p\hat{f}(p) > 0$ and we use the property of $h$-smooth kernel, $q(p) \leq \frac{1}{h}$. Combining the results equation 9, equation 11 and equation 12, we finally bound equation 8:

$$\mathbb{E}_{p\sim P, p^\star\sim Q}\left[p^\star f(p^\star) - pf(p) - \frac{\gamma}{8h}\left(f(p) - \hat{f}(p)\right)^2\right]$$

$$\leq (1 - M(\mathcal{P}))\left(-\hat{p}\hat{f}(\hat{p}) + \frac{h\hat{p}^2}{\gamma}\right) - \hat{p}\hat{f}(\hat{p})M(\mathcal{P}) + \frac{1 - M(\mathcal{P})}{\gamma} + \frac{hM(\mathcal{P})}{\gamma} + \frac{1}{\gamma} + \hat{p}\hat{f}(\hat{p})$$

$$= \frac{(1 - M(\mathcal{P}))h\hat{p}^2}{\gamma} + \frac{hM(\mathcal{P})}{\gamma} + \frac{1}{\gamma} + \frac{1 - M(\mathcal{P})}{\gamma} \leq \frac{2h}{\gamma} + \frac{1}{\gamma} \leq \frac{3}{\gamma}.$$

Since the result holds for any $Q \in \mathcal{Q}_h$ and $f \in \mathcal{F}$, the proof is complete. □

**Lemma A.5.** *Fix $f, \hat{f} \in \mathcal{F}$, $x \in \mathcal{X}$, and $\gamma > 0$. Then for any measures $P, Q$ on $\mathcal{P}$ satisfying $Q \ll P$, the following holds.*

$$\mathbb{E}_{p\sim P, p^\star\sim Q}\left[p^\star f(x, p^\star) - pf(x, p) - \frac{\gamma}{4}\left(f(x, p) - \hat{f}(x, p)\right)^2\right]$$

$$\leq \mathbb{E}_Q\left[p\hat{f}(x, p)\right] - \mathbb{E}_P\left[p\hat{f}(x, p)\right] + \frac{h}{\gamma}\mathbb{E}_P\left[\left(\frac{dQ}{dP} - 1\right)^2\right]$$

*Proof.* Let us simplify the notations as $f(p) := f(x, p)$ and $\hat{f}(p) := \hat{f}(x, p)$. Rearranging the first two terms inside the expectation, we have

$$p^\star f(p^\star) - pf(p) = p^\star\hat{f}(p^\star) - p\hat{f}(p) + p^\star(f(p^\star) - \hat{f}(p^\star)) - p(f(p) - \hat{f}(p))$$

Define $\delta(p) := f(p) - \hat{f}(p)$. Then it follows that

$$\mathbb{E}_{p\sim P, p^\star\sim Q}[p^\star f(p^\star) - pf(p)] = \mathbb{E}_Q\left[p\hat{f}(p)\right] - \mathbb{E}_P\left[p\hat{f}(p)\right] + \mathbb{E}_P\left[\frac{dQ}{dP}p\delta(p) - p\delta(p)\right]. \quad (13)$$

For any $p \in [0, 1]$, we have

$$\frac{dQ}{dP}p\delta(p) - p\delta(p) - \frac{\gamma}{4h}\delta(p)^2 = \frac{hp^2}{\gamma}\left(\frac{dQ}{dP} - 1\right)^2 - \frac{\gamma}{4h}\left(\left(\frac{dQ}{dP} - 1\right)\frac{2hp}{\gamma} - \delta(p)\right)^2$$

$$\leq \frac{hp^2}{\gamma}\left(\frac{dQ}{dP} - 1\right)^2 \leq \frac{h}{\gamma}\left(\frac{dQ}{dP} - 1\right)^2.$$

Combining equation **??** and equation 13, we obtain

$$\mathbb{E}_{p\sim P, p^\star\sim Q}\left[p^\star f(p^\star) - pf(p) - \frac{\gamma}{8h}D_{KL}\left(\text{Ber}(f(x, p))\|\text{Ber}(f^\star(x, p))\right)\right]$$

$$\leq \mathbb{E}_Q\left[p\hat{f}(p)\right] - \mathbb{E}_P\left[p\hat{f}(p)\right] + \frac{h}{\gamma}\mathbb{E}_P\left[\left(\frac{dQ}{dP} - 1\right)^2\right]$$

$$(14)$$

Since equation 14 holds for all $f \in \mathcal{F}$, the proof is complete. □

For the last step, the following lemma bridges Assumption 3.1 and DEC.

**Lemma A.6.** *Suppose Assumption 3.1 holds. If $\ell$ is set to be logarithmic loss $\ell_i(f) = -y_i \log(f(x_i, p_i)) - (1 - y_i) \log(1 - f(x_i, p_i))$ or square loss $\ell_i(f) = (f(x_i, p_i) - y_i)^2$, we have that*

$$\mathbb{E}\left[\sum_{i=1}^{t} (\hat{f}_i(x_i, p_i) - f^\star(x_i, p_i))^2\right] \leq \text{Regret}_R(t)$$

*Proof.* Since $f^\star \in \mathcal{F}$, we have

$$\text{Regret}_R(t) \geq \mathbb{E}\left[\sum_{i=1}^{t} \ell_i(\hat{f}_i) - \inf_{f \in \mathcal{F}} \sum_{i=1}^{t} \ell_i(f)\right] \geq \mathbb{E}\left[\sum_{i=1}^{t} \ell_i(\hat{f}_i) - \sum_{i=1}^{t} \ell_i(f^\star)\right].$$

First consider the case $\ell_i(f) = -y_i \log(f(x_i, p_i)) - (1 - y_i) \log(1 - f(x_i, p_i))$. By Assumption 3.1, we have

$$\mathbb{E}\left[\sum_{i=1}^{t} \ell_i(\hat{f}_i) - \sum_{i=1}^{t} \ell_i(f^\star)\right]$$

$$= \mathbb{E}\left[\sum_{i=1}^{t} \left(y_i \log \frac{f^\star(x_i, p_i)}{\hat{f}_i(x_i, p_i)} + (1 - y_i) \log \frac{1 - f^\star(x_i, p_i)}{1 - \hat{f}_i(x_i, p_i)}\right)\right]$$

$$= \mathbb{E}\left[\sum_{i=1}^{t} \left(f^\star(x_i, p_i) \log \frac{f^\star(x_i, p_i)}{\hat{f}_i(x_i, p_i)} + (1 - f^\star(x_i, p_i)) \log \frac{1 - f^\star(x_i, p_i)}{1 - \hat{f}_i(x_i, p_i)}\right)\right]$$

$$= \mathbb{E}\left[\sum_{i=1}^{t} D_{KL}\left(\text{Ber}(f^\star(x_i, p_i)) \| \text{Ber}(\hat{f}_i(x_i, p_i))\right)\right]$$

$$\geq 2\mathbb{E}\left[\sum_{i=1}^{t} \left(f^\star(x_i, p_i) - \hat{f}_i(x_i, p_i)\right)^2\right].$$

where we use the law of total expectation in the second step and the last step holds due to Pinsker's inequality.

Now consider the case $\ell_i(f) = (f(x_i, p_i) - y_i)^2$. Similarly, we have that

$$\mathbb{E}\left[\sum_{i=1}^{t} \ell_i(\hat{f}_i) - \sum_{i=1}^{t} \ell_i(f^\star)\right]$$

$$= \mathbb{E}\left[\sum_{i=1}^{t} \left\{(\hat{f}_i(x_i, p_i) - y_i)^2 - (f^\star(x_i, p_i) - y_i)^2\right\}\right]$$

$$= \mathbb{E}\left[\sum_{i=1}^{t} (\hat{f}_i(x_i, p_i) - f^\star(x_i, p_i))(\hat{f}_i(x_i, p_i) + f^\star(x_i, p_i) - 2y_i)\right]$$

$$= \mathbb{E}\left[\sum_{i=1}^{t} (\hat{f}_i(x_i, p_i) - f^\star(x_i, p_i))(\hat{f}_i(x_i, p_i) + f^\star(x_i, p_i) - 2f^\star(x_i, p_i))\right]$$

$$= \mathbb{E}\left[\sum_{i=1}^{t} \left(\hat{f}_i(x_i, p_i) - f^\star(x_i, p_i)\right)^2\right].$$

$\square$

Finally, we prove the upper bound based on the supporting lemmas.

**Theorem 5.3 (Restated).** Under Assumption 3.1 and 5.2, setting $\gamma_0 = \Theta((L + 1)^{-\frac{1}{3}})$, Algorithm 1 guarantees that

$$\text{Regret}(T) \leq \tilde{\mathcal{O}}\left(T^{\frac{2}{3}} \cdot \text{Regret}_R(T)^{\frac{1}{3}}\right).$$

*Proof.* The smooth regret of $k$-th episode is decomposed as

$$\text{Regret}_h(\mathcal{E}_k) = \sum_{t \in \mathcal{E}_k} \mathbb{E}\left[\text{Smooth}_h(x_t) - p_t f^\star(x_t, p_t)\right]$$

$$= \underbrace{\sum_{t \in \mathcal{E}_k} \mathbb{E}\left[\text{Smooth}_h(x_t) - p_t f^\star(x_t, p_t) - \frac{\gamma_k}{8h}\left(f^\star(x_t, p_t) - \hat{f}_t(x_t, p_t)\right)^2\right]}_{(I)} \quad (15)$$

$$+ \underbrace{\frac{\gamma_k}{8h}\sum_{t \in \mathcal{E}_k} \mathbb{E}\left[\left(f^\star(x_t, p_t) - \hat{f}_t(x_t, p_t)\right)^2\right]}_{(II)}.$$

By Lemma A.4, it holds that

$$(I) = \mathbb{E}\left[\text{Smooth}_h(x_t) - p_t f^\star(x_t, p_t) - \frac{\gamma_k}{8h}\left(f^\star(x_t, p_t) - \hat{f}_t(x_t, p_t)\right)^2\right]$$

$$= \sup_{Q \in \mathcal{Q}_h} \mathbb{E}_{p_t^\star \sim Q, p_t \sim P_t}\left[p_t^\star f^\star(x_t, p_t^\star) - p_t f^\star(x_t, p_t) - \frac{\gamma_k}{8h}\left(f^\star(x_t, p_t) - \hat{f}_t(x_t, p_t)\right)^2\right] \le \frac{3}{\gamma_k}.$$

Since the $k$-th episode is $\mathcal{E}_k = \{\sum_{r=1}^{k-1} \tau_r + 1, \ldots, \sum_{r=1}^{k} \tau_r\}$ with $\tau_k = 2^k$, thus $\sum_{r=1}^{k} \tau_r = 2\tau_k - 2$. Using this fact with Lemma A.6, we obtain

$$(II) \le \frac{\gamma_k}{8h}\mathbb{E}\left[\sum_{i=1}^{\lceil 2\tau_k - 2 \rceil}\left(f^\star(x_t, p_t) - \hat{f}_t(x_t, p_t)\right)^2\right] \le \frac{\gamma_k}{8h}\text{Regret}_R(2\tau_k - 2). \quad (16)$$

Here, without loss of generality, we assume that $T = \sum_{k=1}^{N_\mathcal{E}} \tau_k$ where $N_\mathcal{E}$ is the number of episodes. Therefore, the smooth regret from the $k$-th episode is bounded by

$$\text{Regret}_h(\mathcal{E}_k) \le \frac{3\tau_k}{\gamma_k} + \frac{\gamma_k}{8h}\text{Regret}_R(2\tau_k - 2).$$

Furthermore, Lemma A.2 implies that the standard regret within $\mathcal{E}_k$ is bounded as follows:

$$\text{Regret}(\mathcal{E}_k) \le \text{Regret}_h(\mathcal{E}_k) + (L+1)h\tau_k \le \frac{3\tau_k}{\gamma_k} + \frac{\gamma_k}{8h}\text{Regret}_R(2\tau_k - 2) + (L+1)h\tau_k.$$

Setting the parameters as

$$h = \Theta((L+1)^{-2/3}\tau_k^{-1/3}\text{Regret}_R(2\tau_k - 2)^{1/3}) \quad \text{and}$$

$$\gamma_k = \Theta((L+1)^{-1/3}\tau_k^{1/3}\text{Regret}_R(2\tau_k - 2)^{-1/3}),$$

we have

$$\text{Regret}(\mathcal{E}_k) \le \Theta\left((L+1)^{1/3}\tau_k^{2/3}\text{Regret}_R(2\tau_k - 2)^{1/3}\right)$$

Taking a summation of all episodes, the standard regret for $T$ steps is bounded by

$$\text{Regret}(T) = \sum_{k=1}^{N_\mathcal{E}} \text{Regret}(\mathcal{E}_k) \lesssim \text{Regret}_R(T)^{1/3}\sum_{k=1}^{N_\mathcal{E}} 2^{\frac{2k}{3}} \le \tilde{\mathcal{O}}\left(T^{2/3}\text{Regret}_R(T)^{1/3}\right)$$

where the last step holds due to the fact that $\sum_{k=1}^{N_\mathcal{E}} 2^{\frac{2k}{3}} = (2^{\frac{2N_\mathcal{E}}{3}} - 1)/(2^{2/3} - 1) \lesssim T^{2/3}$ and $\lesssim$ hides absolute constants. This completes the proof. □

# B `DP-IGW` WITH OFFLINE REGRESSION ORACLE

In this section, we provide an alternative offline regression oracle-based Algorithm 2 and show its performance guarantees. Suppose the offline regression oracle $\text{Alg}_R$ satisfies Assumption B.1, then Algorithm 2 guarantees bounded regret as Theorem B.2. $\text{Alg}_R$ is updated every epoch unlike 1 where $\text{Alg}_R$ is updated every time step, therefore computationally efficient.

---

**Algorithm 2** DP-IGW with offline regression oracle

---

1: **Input:** Regression oracle $\text{Alg}_R$, epoch schedule $\{\tau_k\}_{k=1}^{k(T)}$, price discretization number $K$, confidence parameter $\delta$, tuning parameter $c$
2: **for** Epoch $k = 1, 2, \ldots$ **do**
3:    Set $\gamma_k = c\sqrt{K/\mathcal{E}_{\mathcal{F}, \delta/(2k^2)}(\tau_{k-1} - \tau_{k-2})}$
4:    Update $\hat{f}_k$ with $\{(x_i, p_i, y_i)\}_{i=\tau_{k-2}+1}^{\tau_{k-1}}$ via $\text{Alg}_R$
5:    **for** $t$ in epoch $k$ **do**
6:       Observe $x_t$ and access $\hat{f}_k(x_t, \cdot)$
7:       Compute $\hat{p}_t \leftarrow \arg\max_{p \in \mathcal{P}} p\hat{f}_t(x_t, p)$, then sample $p_t \sim P_k(\cdot|x_t)$, where

$$P_k(p|x_t) = \begin{cases} \frac{1}{K + \gamma_k(\hat{f}_k(x_t, \hat{p}_t) - \hat{f}_k(x_t, p))} & \text{for } p \neq \hat{p}_t \\ 1 - \sum_{p \neq \hat{p}_t} p_t(p) & \text{for } p = \hat{p}_t \end{cases}$$

8:       Set price $p_t$ and observe $y_t$
9:    **end for**
10: **end for**

---

**Assumption B.1.** *(Offline regression oracle) The regression oracle $\text{Alg}_R$ guarantees that given i.i.d. sampled history $\mathcal{H}_t := \{(x_i, p_i, y_i)\}_{i=1}^n$ according to $x_i \sim \mathcal{D}_x, p_i \sim P(\cdot|x_i)$ where $P$ is an arbitrary action selection kernel, $\ell$ is the loss function, either logarithmic loss $\ell(f) = -y\log(f(x, p)) - (1 - y)\log(1 - f(x, p))$ or square loss $\ell(f) = (f(x, p) - y)^2$, with probability at least $1 - \delta$, the expected estimation is bounded by the offline learning guarantee $\mathcal{E}_{\mathcal{F}, \delta}(n)$ that decreases with $n$:*

$$\mathbb{E}\left[\ell(\hat{f}) - \inf_{f \in \mathcal{F}} \ell(f)\right] \leq \mathcal{E}_{\mathcal{F}, \delta}(n).$$

**Theorem B.2.** *Under Assumption B.1 and Assumption 5.1, setting $\tau_k = 2^k, c = 1/2$, Algorithm 2 guarantees that*

$$\text{Regret}(T) \leq \mathcal{O}\left(T \cdot (\mathcal{E}_{\mathcal{F}, \delta/\log T}(T))^{1/3}\right).$$

*Proof.* Let $K$ be the discretization number of the price space. Setting $\tau_k = 2^k$, we have the regret bounded by Lemma B.7 summed with cumulative gap due to discretization as the following,

$$\text{Regret}(T) \leq \mathcal{O}\left(\sqrt{K\mathcal{E}_{\mathcal{F}, \delta/\log T}(T)T}\right) + \mathcal{O}\left(T/K\right).$$

Setting $K = \mathcal{O}\left(\mathcal{E}_{\mathcal{F}, \delta/\log T}(T)^{-1/3}\right)$ which minimizes RHS, we get

$$\text{Regret}(T) \leq \mathcal{O}\left(T \cdot (\mathcal{E}_{\mathcal{F}, \delta/\log T}(T))^{1/3}\right).$$

$\square$

Our proof is inspired by Simchi-Levi & Xu (2022), and we show analogous lemmas. Define:

$$V(P, P') = \mathbb{E}_{x \sim \mathcal{D}_x, p \sim P'(\cdot|x)}\left[\frac{1}{P(p|x)}\right], V_t(P) = \max_{1 \leq k \leq k(t)-1} V(P_k, P)$$

$$\mathcal{R}(P) := \mathbb{E}_{x \sim \mathcal{D}_x, p \sim P(\cdot|x)}[pf^*(x, p)], \hat{\mathcal{R}}_t(P) := \mathbb{E}_{x \sim \mathcal{D}_x, p \sim P(\cdot|x)}[p\hat{f}_{k(t)}(x, p)]$$

$$\text{Regret}(P) := \mathcal{R}(p_{f^*}) - \mathcal{R}(P), \widehat{\text{Regret}}_t(P) := \hat{\mathcal{R}}_t(p_{\hat{f}_{k(t)}}) - \hat{\mathcal{R}}_t(P).$$

We consider logarithmic loss $\ell(f) = -y\log(f(x, p)) - (1 - y)\log(1 - f(x, p))$ or square loss $\ell(f) = (f(x, p) - y)^2$.

**Lemma B.3.** *(Analogous to Lemma A.2 in Simchi-Levi & Xu (2022))*
*With probability at least $1 - \delta/2$, $\forall k \geq 2$ and $t$ in epoch $k$, $\mathbb{E}[(\hat{f}_k(x_t, p_t) - f^*(x_t, p_t))^2] \leq K/(4\gamma_k^2)$.*

*Proof.* Fix epoch $k$, then for all $t$ in epoch $k$, we show that the following holds with probability at least $1 - \delta/(2k^2)$ for both logarithmic loss and square loss,

$$\mathbb{E}\left[(\hat{f}_k(x_t, p_t) - f^*(x_t, p_t))^2\right] \leq \mathbb{E}\left[\ell(\hat{f}_k) - \inf_{f \in \mathcal{F}} \ell(f)\right] \leq \mathcal{E}_{\mathcal{F}, \delta/(2k^2)}(\tau_{k-1} - \tau_{k-2}) = \frac{K}{4\gamma_k^2}.$$

First consider the case $\ell(f) = -y \log(f(x, p)) - (1 - y) \log(1 - f(x, p))$. By Assumption B.1,

$$2\mathbb{E}\left[(\hat{f}_k(x_t, p_t) - f^*(x_t, p_t))^2\right] \leq \mathbb{E}\left[D_{KL}\left(\text{Ber}(\hat{f}_k(x_t, p_t) \| \text{Ber}(f^*(x_t, p_t)))\right)\right]$$

$$= \mathbb{E}\left[f^*(x_t, p_t) \log \frac{f^*(x_t, p_t)}{\hat{f}_k(x_t, p_t)} + (1 - f^*(x_t, p_t)) \log \frac{1 - f^*(x_t, p_t)}{1 - \hat{f}_k(x_t, p_t)}\right]$$

$$= \mathbb{E}\left[y_t \log f^*(x_t, p_t) + (1 - y_t) \log(1 - f^*(x_t, p_t)) - y_t \log \hat{f}_k(x_t, p_t) - (1 - y_t)(1 - \hat{f}_k(x_t, p_t))\right]$$

$$= \mathbb{E}\left[\ell(\hat{f}_k(x_t, p_t), y_t) - \ell(f^*(x_t, p_t), y_t)\right] \leq \mathbb{E}\left[\ell(\hat{f}_k) - \ell(f^*)\right] \leq \mathbb{E}\left[\ell(\hat{f}_k) - \inf_{f \in \mathcal{F}} \ell(f)\right].$$

where the first inequality is due to Pinsker's inequality and the last inequality is because $f^* \in \mathcal{F}$. Apply union bound. Now consider the case $\ell(f) = (f(x, p) - y)^2$. Similarly, we have that

$$\mathbb{E}\left[(\hat{f}_k(x_t, p_t) - f^*(x_t, p_t))^2\right] = \mathbb{E}\left[(\hat{f}_k(x, p) - f^*(x, p))(\hat{f}_k(x, p) + f^*(x, p) - 2f^*(x, p))\right]$$

$$= \mathbb{E}\left[(\hat{f}_k(x, p) - f^*(x, p))(\hat{f}_k(x, p) + f^*(x, p) - 2y)\right] = \mathbb{E}\left[(\hat{f}_k(x, p) - y)^2 - (f^*(x, p) - y)^2\right]$$

$$= \mathbb{E}\left[\ell(\hat{f}_k) - \ell(f^*)\right] \leq \mathbb{E}\left[\ell(\hat{f}_k) - \inf_{f \in \mathcal{F}} \ell(f)\right].$$

Therefore we have for all $t$ in each epoch $k$ with probability at least $\delta/(2k^2)$,

$$\mathbb{E}\left[(\hat{f}_k(x_t, p_t) - f^*(x_t, p_t))^2\right] \leq \frac{K}{4\gamma_k^2}.$$

Now apply union bound. $\qquad\square$

**Lemma B.4.** *(Analogous to Lemma A.5, A.6 in Simchi-Levi & Xu (2022))*
*For the algorithm's randomized policy $P_k$ and any randomized policy $P$,*

$$\widehat{\text{Regret}}_t(P_k) \leq K/\gamma_k$$

$$V(P_k, P) \leq K + \gamma_k \widehat{\text{Regret}}_t(P)$$

*Proof.*

$$\widehat{\text{Regret}}_t(P_k) = \mathbb{E}_{x \sim \mathcal{D}_x, p \sim P_k(\cdot|x)}\left[\hat{p}_k \hat{f}_k(x, \hat{p}_k) - p \hat{f}_k(x, p)\right]$$

$$= \mathbb{E}_{x \sim \mathcal{D}_x}\left[\sum_p P_k(p|x)(\hat{p}_k \hat{f}_k(x, \hat{p}_k) - p \hat{f}_k(x, p))\right]$$

$$= \mathbb{E}_{x \sim \mathcal{D}_x}\left[\sum_{p \neq \hat{p}_k} \frac{\hat{p}_k \hat{f}_k(x, \hat{p}_k) - p \hat{f}_k(x, p))}{K + \gamma_k(\hat{p}_k \hat{f}_k(x, \hat{p}_k) - p \hat{f}_k(x, p)))}\right] \leq \frac{K - 1}{\gamma_k}$$

If $p \neq \hat{p}_k$, $1/P_k(p, x) = K + \gamma_k(\hat{p}_k \hat{f}_k(x, \hat{p}_k) - p \hat{f}_k(x, p))$, while if $p = \hat{p}_k$, $1/P_k(p, x) \leq 1/(1 - \frac{K-1}{K}) = K = K + \gamma_k(\hat{p}_k \hat{f}_k(x, \hat{p}_k) - p \hat{f}_k(x, p))$. Therefore,

$$\mathbb{E}_{x \sim \mathcal{D}_x, p \sim P(\cdot|x)}\left[\frac{1}{P_k(p|x)}\right] \leq K + \gamma_k \mathbb{E}_{x \sim \mathcal{D}_x, p \sim P(\cdot|x)}\left[\hat{p}_k \hat{f}_k(x, \hat{p}_k) - p \hat{f}_k(x, p)\right] = K + \gamma_k \widehat{\text{Regret}}_t(P)$$

$$\square$$

**Lemma B.5.** *(Analogous to Lemma A.11 in Simchi-Levi & Xu (2022))*
*For any randomized policy P,*

$$|\hat{\mathcal{R}}_t(P) - \mathcal{R}(P)| \leq \frac{\sqrt{V_t(P)}\sqrt{K}}{2\gamma_{k(t)}}$$

*Proof.* Let $s_0 = \tau_{k(t)-2} + 1$.

$$V_t(P) \sum_{s=s_0}^{\tau_{k(t)-1}} \mathbb{E}\left[(\hat{f}_k(x_s, p_s) - f^*(x_s, p_s))^2\right] \geq \sum_{s=s_0}^{\tau_{k(t)-1}} V(P_k, P)\mathbb{E}\left[(\hat{f}_k(x_s, p_s) - f^*(x_s, p_s))^2\right]$$

$$= \sum_{s=s_0}^{\tau_{k(t)-1}} \mathbb{E}_{x_s \sim \mathcal{D}_x, p_s \sim P(\cdot|x_s)}\left[\frac{1}{P_k(p_s|x_s)}\right] \mathbb{E}_{x_s \sim \mathcal{D}_x, p_s \sim P_k(\cdot|x_s)}\left[(\hat{f}_k(x_s, p_s) - f^*(x_s, p_s))^2\right]$$

$$\geq \sum_{s=s_0}^{\tau_{k(t)-1}} \left(\mathbb{E}_{x_s \sim \mathcal{D}_x}\left[\sqrt{\mathbb{E}_{p_s \sim P(\cdot|x_s)}\left[\frac{1}{P_k(p_s|x_s)}\right] \mathbb{E}_{p_s \sim P_k(\cdot|x_s)}\left[(\hat{f}_k(x_s, p_s) - f^*(x_s, p_s))^2\right]}\right]\right)^2$$

$$= \sum_{s=s_0}^{\tau_{k(t)-1}} \left(\mathbb{E}_{x_s \sim \mathcal{D}_x}\left[\sqrt{\sum_p \frac{P(p|x_s)}{P_k(p|x_s)} \sum_p P_k(p|x_s)(\hat{f}_k(x_s, p) - f^*(x_s, p))^2}\right]\right)^2$$

$$\geq \sum_{s=s_0}^{\tau_{k(t)-1}} \left(\mathbb{E}_{x_s \sim \mathcal{D}_x}\left[\left|\sum_p \sqrt{P(p|x_s)}\left|\hat{f}_k(x_s, p) - f^*(x_s, p)\right|\right|\right]\right)^2$$

$$\geq \sum_{s=s_0}^{\tau_{k(t)-1}} \left(\mathbb{E}_{x_s \sim \mathcal{D}_x}\left[\sum_p P(p|x_s)\left|\hat{f}_k(x_s, p) - f^*(x_s, p)\right|\right]\right)^2$$

$$= \sum_{s=s_0}^{\tau_{k(t)-1}} \left(\mathbb{E}_{x_s \sim \mathcal{D}_x}\left[\mathbb{E}_{p_s \sim P(\cdot|x_s)}\left[\left|\hat{f}_k(x_s, p_s) - f^*(x_s, p_s)\right|\right]\right]\right)^2$$

$$\geq \sum_{s=s_0}^{\tau_{k(t)-1}} \left(\mathbb{E}_{x_s \sim \mathcal{D}_x}\left[\mathbb{E}_{p_s \sim P(\cdot|x_s)}\left[|p_s|\left|\hat{f}_k(x_s, p_s) - f^*(x_s, p_s)\right|\right]\right]\right)^2$$

$$\geq \sum_{s=s_0}^{\tau_{k(t)-1}} |\hat{\mathcal{R}}_t(P) - \mathcal{R}(P)|^2 = (\tau_{k(t)-1} - s_0 + 1)|\hat{\mathcal{R}}_t(P) - \mathcal{R}(P)|^2$$

where we use Cauchy-Schwarz inequality, $0 \leq P(p|x_s) \leq 1$, $0 \leq p_s \leq 1$, and convexity of L1-norm. Therefore we have

$$|\hat{\mathcal{R}}_t(P) - \mathcal{R}(P)| \leq \sqrt{V_t(P)}\sqrt{\frac{\sum_{s=\tau_{k(t)-2}+1}^{\tau_{k(t)-1}} \mathbb{E}\left[(\hat{f}_k(x_s, p_s) - f^*(x_s, p_s))^2\right]}{\tau_{k(t)-1} - \tau_{k(t)-2}}} \leq \frac{\sqrt{V_t(P)}\sqrt{K}}{2\gamma_{k(t)}}$$

where the final inequality follows from Lemma B.3. □

**Lemma B.6.** *(Analogous to Lemma A.12 in Simchi-Levi & Xu (2022))*
*Let $c_0 = 5.15$. For all epochs $k$, all rounds $t$ in epoch $k$, and randomized policy $P$,*

$$\text{Regret}(P) \leq 2\widehat{\text{Regret}}_t(P) + c_0 K/\gamma_k$$

$$\widehat{\text{Regret}}_t(P) \leq 2\text{Regret}(P) + c_0 K/\gamma_k$$

*Proof.* The proof follows directly from Lemma A.12 in Simchi-Levi & Xu (2022). □

**Lemma B.7.** *(Analogous to Theorem 2 in Simchi-Levi & Xu (2022)*
*With epoch schedule $\tau_k \geq 2^k$, $c = 1/2$, Algorithm (to be stated in appendix) guarantees that for any $T$ with probability at least $1 - \delta$,*

$$\text{Regret}(T) \leq \mathcal{O}\left(\sqrt{K} \sum_{k=2}^{k(T)} \sqrt{\mathcal{E}_{\mathcal{F}, \delta/(2k)^2}(\tau_{k-1} - \tau_{k-2})}(\tau_{k-1} - \tau_{k-2})\right).$$

*Proof.* The proof follows directly from the proof of Theorem 2 in Simchi-Levi & Xu (2022). $\quad\square$

## C   PROOF OF THEOREM 5.5

**Theorem 5.5 (Restated).** Under Assumption 5.4, any dynamic pricing algorithm has regret

$$\text{Regret}(T) \geq \Omega(T^{\frac{d+2}{d+3}}).$$

If the reference function class is finite, we have

$$\text{Regret}(T) \geq \Omega(T^{2/3} \log^{1/3}(|\mathcal{F}|)).$$

*Proof.* Let $S_x$ be a $\epsilon$-net of $\mathcal{X}$, with $n_x = |S_x|$. Let $S_p$ be a $\epsilon$-net of $[\frac{1}{3}, \frac{2}{3}]$, a subset of the price space, and denote $n_p = |S_p|$. Then we have $n_x = \Theta(\epsilon^{-d})$ and $n_p = \Theta(\epsilon^{-1})$. We construct a collection of functions as follows: For each $x_0 \in S_x$, randomly and independently choose $p_0 \in S_p$ and set $f(x_0, p_0) = \frac{C\epsilon}{4}$ for some constant $C$ (the value will be specified later) and set $f(x_0, \tilde{p}_0) = 0$ for all other $\tilde{p}_0 \in S_p$. Define

$$f(x, p) := \max_{(x_0, p_0) \in S_x \times S_p} \max\{0, f(x_0, p_0) - C(\|x - x_0\|_2 + |p - p_0|)\}.$$

which makes $f$ $C$-Lipschitz. Since the choice of $p_0$ is independent of $x_0$, the feedback under some context $x_0$ reveals no information about the values $f(\tilde{x}_0, \cdot), \tilde{x}_0 \in S_x \setminus \{x_0\}$.

Suppose the sequence $\{x_t\}_{t=1}^T$ is a repeated permutation of the set $S_x$. Let $I(x_0) = \{t \in [T] : x_t = x_0\}$ for each $x_0 \in S_x$. We can decomposed the regret as $\text{Regret}(T) = \sum_{x_0 \in S_x} \text{Regret}_{x_0}(T)$ where $\text{Regret}_{x_0}(T) = \mathbb{E}[\sum_{t \in I(x_0)} f(x_0, p_t^\star) - f(x_0, p_t)]$. Since feedbacks from $[T] \setminus I(x_0)$ reveals no information about $f(x_0, \cdot)$, any algorithm $\mathcal{A}$ induces $n_x$ sub algorithms $\mathcal{A}_{x_0}$ for each $x_0 \in S_x$, and $\mathcal{A}$ simulates $\mathcal{A}_{x_0}$ in $I(x_0)$. Then we have $\mathbb{E}[\text{Regret}_{x_0}(T)] = \mathbb{E}[\text{Regret}_{\mathcal{A}_{x_0}}(T/n_x)]$ where the expectations are taken over possible problem instances. In what follows, we fix $x_0$ and prove a lower bound of the contextless dynamic pricing problem associated with $x_0$.

Simplifying the notation $f(p) := f(x_0, p)$, $f(p)$ is $C$-Lipschitz, unimodal, and differentiable on $[0, 1]$ except at most 3 points. We denote $S_d := \{p \in [0, 1] : f \text{ is differentiable at } p\}$. Define $g(p) = 1 - \frac{1}{1+f(p)}$ which is $C$-Lipschitz since

$$|g(p) - g(p')| = \left| \frac{f(p) - f(p')}{(1 + f(p))(1 + f(p'))} \right| \leq |f(p) - f(p')|.$$

Also, $g(p)$ is differentiable on the set where $f(p)$ is differentiable, and $|g'(p)| \leq C$ on the set.

Assume $C < 1$, let $b = \frac{1+C}{2} \in (0, 1)$ and define $F(p)$:

$$F(p) = \begin{cases} 0 & 0 \leq p \leq b \\ 1 - \frac{b}{p} - \frac{1-b}{p} g\left(\frac{p-b}{1-b}\right) & b < p \leq 1 \end{cases}$$

$F(p)$ has key properties to define a problem instance.

1. $F(p)$ is non-decreasing.

   *Proof.* It is trivial on $[0, b]$. On $(b, 1) \cap S_b$, $F'(p) = \frac{1}{p^2}\left(b - pg'\left(\frac{p-b}{1-b}\right) + (1-b)g\left(\frac{p-b}{1-b}\right)\right)$.

   Since $b - pg'\left(\frac{p-b}{1-b}\right) \geq b - 1 \cdot C > 0$, $F(p)$ is non-decreasing on $(b, 1) \cap S_b$. Since $F$ is continuous on $[0, 1]$ and differentiable except finitely many points, $F$ is non-decreasing on $[0, 1]$. $\quad\square$

2. $F(p)$ is Lipschitz continuous.

*Proof.* By definition $F(p)$ is constant on $[0, b]$. On $(b, 1) \cap S_b$, $|F'(p)| = \left| \frac{1}{p^2} \left( b - pg' \left( \frac{p-b}{1-b} \right) + (1-b)g \left( \frac{p-b}{1-b} \right) \right) \right|$

$\leq \left| \frac{b + C + (1-b)}{b^2} \right| \leq 4$. By triangular inequality, $F(p)$ is 12-Lipschitz on $[0, 1]$. $\qquad \square$

3. There exists a unique maximizer for the revenue function $r(p) = p(1 - F(p))$.

*Proof.* By the definition of $F(p)$, we have

$$p(1 - F(p)) = \begin{cases} p & 0 \leq p \leq b \\ b + (1-b)g \left( \frac{p-b}{1-b} \right) & b < p \leq 1 \end{cases}$$

hence $p(1 - F(p)) \leq b$ on $[0, b]$ and $p(1 - F(p)) \geq b$ on $(b, 1]$. Since $g(p)$ has the same unique maximizer $p^*$ with $f(p)$, $p(1 - F(p))$ also has the unique maximizer $b + (1-b)p^* \in [0, 1]$. $\qquad \square$

Since $F(p)$ is non-decreasing in $p$ and Lipschitz continuous, we can associate a problem instance for each $F(p)$. Specifically, $1 - F(p)$ is the probability of purchase conditioned on the price $p$, i.e. $P(y_t = 1 \mid p_t = p) = 1 - F(p)$. Note that there exists a bijection from $p_0 \in S_p$ to $F$. In the following parts, we use $\{F_j\}_{j=1}^{n_p}$ to denote the set of functions generated by $S_p = \{p_j : j \in n_p\}$, and define $F_0(p)$ be the function derived by $f_0(p) := 0$. Without loss of generality, we assume $p_1 < p_2 < \cdots < p_{n_p}$. Given a policy $\pi$, denote the probability distribution over trajectories $u_t = (p_1, y_1, p_2, y_2, \ldots, p_t, y_t)$, determined by $\pi$ and $F_j$, by $\mathbb{P}_j$. Since for any $F_j$, price $p \in [0, b)$ incurs regret greater than that incurred by $p \in [b, 1]$, we assume $p_t \in [b, 1]$ for all $t$. Further, define $U_j = \left[ \frac{p_{j-1} + p_j}{2}, \frac{p_j + p_{j+1}}{2} \right)$ for $j \in \{2, \ldots, n_p - 1\}$, $U_1 = \left[ \frac{1}{3}, \frac{p_1 + p_2}{2} \right)$, $U_{n_p} = \left[ \frac{p_{n_p-1} + p_{n_p}}{2}, \frac{2}{3} \right]$.

Now we bound the KL divergence $D_{KL}(\mathbb{P}_0 \| \mathbb{P}_j)$ for any $j \in [n_p]$. We have

$$D_{KL}(\mathbb{P}_0 \| \mathbb{P}_j) = \mathbb{E}_{\mathbb{P}_0} \left[ \log \frac{\mathbb{P}_0(u_t)}{\mathbb{P}_j(u_t)} \right]$$

$$= \mathbb{E}_{\mathbb{P}_0} \left[ \log \frac{\Pi_{i=1}^t \pi(p_i \mid p_1, \ldots, y_{i-1}) P_0(y_i \mid p_i)}{\Pi_{i=1}^t \pi(p_i \mid p_1, \ldots, y_{i-1}) P_j(y_i \mid p_i)} \right]$$

$$= \mathbb{E}_{\mathbb{P}_0} \left[ \log \frac{\Pi_{i=1}^t P_0(y_i \mid p_i)}{\Pi_{i=1}^t P_j(y_i \mid p_i)} \right] = \mathbb{E}_{\mathbb{P}_0} \left[ \sum_{i=1}^t \log \frac{P_0(y_i \mid p_i)}{P_j(y_i \mid p_i)} \right]$$

$$= \sum_{i=1}^t \mathbb{E}_{\mathbb{P}_0} \left[ D_{KL}(P_0(\cdot \mid p_i) \| P_j(\cdot \mid p_i)) \right]$$

$$= \sum_{i=1}^t \mathbb{E}_{\mathbb{P}_0} \left[ D_{KL}(\text{Ber}(1 - F_0(p_i)) \| \text{Ber}(1 - F_j(p_i))) \right]$$

$$= \sum_{i=1}^t \mathbb{E}_{\mathbb{P}_0} \left[ \mathbb{1}\{ \frac{p_i - b}{1 - b} \in U_j \} D_{KL}(\text{Ber}(1 - F_0(p_i)) \| \text{Ber}(1 - F_j(p_i))) \right]$$

where the last step holds due to the fact that $F_0(p) = F_j(p)$ for $\frac{p-b}{1-b} \notin U_j$. For the range of the Bernoulli parameters, since $p \in [b, 1]$ and $g_0(p) \geq 0$, we have $1 - F_0(p_i) = \frac{b + (1-b)g_0(p_i)}{p_i} \geq b \geq \frac{1}{2}$. Moreover, due to the fact that $F_j$ is non-decreasing and $\frac{p_i - b}{1 - b} \in U_j \Rightarrow p_i \geq b + \frac{1-b}{3}$ implies

$$1 - F_j(p_t) \leq 1 - F_j(b + \frac{1-b}{3}) \leq \frac{b + (1-b) \cdot g_j(1/3)}{b + \frac{1-b}{3}}.$$

Setting $C = \frac{1}{16}$ and $b = \frac{17}{32}$ while assuming $\epsilon \leq 1$, we have $1 - F_j(p_i) \leq \frac{5}{6}$. Hence, for all $\frac{p_i - b}{1 - b} \in U_j$, it holds that

$$\frac{1}{2} \leq 1 - F_0(p_i) \leq 1 - F_j(p_i) \leq \frac{1}{2} + \frac{1}{3}$$

where the second inequality holds by the definition of $F_0$. Therefore, by Lemma C.1, it holds that

$$D_{KL}\left(\text{Ber}(1-F_0(p_i))\|\text{Ber}(1-F_j(p_i))\right) \leq \frac{4}{1-4\cdot(\frac{1}{3})^2}((1-F_0(p_i))-(1-F_j(p_i)))^2$$

$$= \frac{36}{5}\left(\frac{(1-b)(g_0(p_i)-g_j(p_i))}{p_i}\right)^2$$

$$= \frac{36}{5}\left(\frac{(1-b)(f_0(p_i)-f_j(p_i))}{(1+f_0(p_i))(1+f_j(p_i))}\right)^2 \leq \frac{576}{5}\epsilon^2$$

On the other hand, using Lemma C.2 with $h(u_t) = N_j(u_t) = |\{i : \frac{p_i-b}{1-b} \in U_j, i \in [t]\}|$, since $0 \leq N_j \leq t$, we have

$$\mathbb{E}_{\mathbb{P}_j}[N_j] - \mathbb{E}_{\mathbb{P}_0}[N_j] \leq t\sqrt{\frac{1}{2}D_{KL}(\mathbb{P}_0\|\mathbb{P}_j)}$$

$$\leq t\sqrt{\frac{1}{2}\sum_{i=1}^{t}\mathbb{E}_{\mathbb{P}_0}\left[\mathbb{1}\left\{\frac{p_i-b}{1-b}\in U_j\right\}D_{KL}\left(\text{Ber}(1-F_0(p_i))\|\text{Ber}(1-F_j(p_i))\right)\right]}$$

$$= t\sqrt{\frac{1}{2}\frac{576}{5}\epsilon^2\sum_{i=1}^{t}\mathbb{P}\left(\frac{p_i-b}{1-b}\in U_j\right)} = t\sqrt{\frac{288}{5}\epsilon^2\mathbb{E}_{\mathbb{P}_0}[N_j]}.$$

Taking summation over all $j \in [n_p]$, we obtain

$$\frac{1}{n_p}\sum_{j=1}^{n_p}\mathbb{E}_{\mathbb{P}_j}[N_j] \leq \frac{1}{n_p}\sum_{j=1}^{n_p}\mathbb{E}_{\mathbb{P}_0}[N_j] + \sum_{j=1}^{n_p}\frac{t}{n_p}\sqrt{\frac{288}{5}\epsilon^2\mathbb{E}_{\mathbb{P}_0}[N_j]} = \frac{t}{n_p} + \frac{t\epsilon}{n_p}\sum_{j=1}^{n_p}\sqrt{\frac{288}{5}\mathbb{E}_{\mathbb{P}_0}[N_j]}$$

$$\leq \frac{t}{n_p} + \frac{t\epsilon}{n_p}\sqrt{\frac{288}{5}n_p\cdot\sum_{j=1}^{n_p}\mathbb{E}_{\mathbb{P}_0}[N_j]} \leq t\cdot\left(\frac{1}{n_p} + \frac{\epsilon}{n_p}\sqrt{\frac{288}{5}n_p t}\right),$$

where we used the fact $\sum_{j=1}^{n_p}\mathbb{E}_{\mathbb{P}_0}[N_j] = t$. Since $n_p = \Theta(\epsilon^{-1})$, for large enough $t$, there exists an absolute constant $c$ such that $\frac{1}{n_p}\sum_{j=1}^{n_p}\mathbb{E}_{\mathbb{P}_j}[N_j] \leq \frac{1}{2}t$ hold, given $t\epsilon^3 = c$. Therefore, there exists some index $j$ such that $\mathbb{E}_{\mathbb{P}_j}[N_j] \leq \frac{1}{2}t$. For such $j$, we can derive a lower bound of regret as follows:

$$\text{Regret}(t) = \mathbb{E}_{\mathbb{P}_j}\left[\sum_{i=1}^{t}(r_j(p_j^*)-r_j(p_i))\right] \geq \mathbb{E}_{\mathbb{P}_j}\left[\sum_{i=1}^{t}\mathbb{1}\{\frac{p_i-b}{1-b}\in U_j\}(r_j(p_j^*)-r_j(p_i))\right]$$

$$= \mathbb{E}_{\mathbb{P}_j}\left[\sum_{i=1}^{t}\mathbb{1}\{\frac{p_i-b}{1-b}\notin U_j\}\frac{(1-b)(f_j(p_j^*)-f_j(p_i))}{(1+f_j(p_j^*))(1+f_j(p_i))}\right] \geq \mathbb{E}_{\mathbb{P}_j}\left[\sum_{i=1}^{t}\mathbb{1}\{\frac{p_i-b}{1-b}\notin U_j\}\frac{\epsilon}{16}\right]$$

$$\geq \frac{\epsilon}{16}\left(t-\mathbb{E}_{\mathbb{P}_j}[N_j]\right) \geq \frac{\epsilon}{16}\frac{t}{2}$$

Now back to the original contextual pricing problem. For each $x_0 \in S_x$, $\mathcal{A}_{x_0}$ is executed for $t = \frac{T}{n_x}$ steps. If $\epsilon$ satisfies $\frac{T}{n_x}\epsilon^3 = \Theta(\epsilon^{d+3}T) = c$, $\mathcal{A}_{x_0}$ incurs at least $\Theta(\epsilon\frac{T}{n_x})$ regret. Therefore, the regret of $\mathcal{A}$ is at least

$$n_x\cdot\Theta\left(\epsilon\frac{T}{n_x}\right) = \Theta\left(\epsilon\cdot T\right) = \Theta\left(T^{\frac{d+2}{d+3}}\right).$$

This proves the first result in Theorem 5.5. For the second result, note that the function class we constructed has cardinality $\log|\mathcal{F}| = n_x\log n_p = \Theta(\epsilon^{-d}\log\frac{1}{\epsilon})$. By the choice of $\epsilon = \Theta(T^{-\frac{1}{d+3}})$, we have $T^{\frac{d+2}{d+3}} = \Theta(T^{2/3}\log^{1/3}|\mathcal{F}|)$.

$\square$

**Lemma C.1** (Lemma 6 in Luo et al. (2022)). *For Bernoulli distributions Ber(p) and Ber(p + ε) with* $1/2 \leq p \leq p + \epsilon \leq 1/2 + C$, *we have*

$$D_{KL}\left(Ber(p)\|Ber(p+\epsilon)\right) \leq \frac{4}{1 - 4C^2}\epsilon^2$$

**Lemma C.2.** *Assume* $\mathbb{P}_1$ *and* $\mathbb{P}_2$ *are distributions over trajectories* $u_t = (p_1, y_1, \ldots, p_t, y_t)$. *For any function* $h$ *on the trajectories that has a bounded value* $[0, M]$, *it holds that*

$$\mathbb{E}_{\mathbb{P}_1}[h(u_t)] - \mathbb{E}_{\mathbb{P}_2}[h(u_t)] \leq M\sqrt{\frac{1}{2}D_{KL}\left(\mathbb{P}_2\|\mathbb{P}_1\right)}. \tag{17}$$

*Proof.* The proof is based on the standard KL divergence argument (Auer et al., 2002; Luo et al., 2022; Xu & Wang, 2022). Consider the measure $\mathbb{Q} = \frac{1}{2}(\mathbb{P}_1 + \mathbb{P}_2)$. Then $\mathbb{P}_1 \ll \mathbb{Q}$ and $\mathbb{P}_2 \ll \mathbb{Q}$, thus the Radon-Nikodym derivatives $\frac{d\mathbb{P}_1}{d\mathbb{Q}} = m_1$ and $\frac{d\mathbb{P}_2}{d\mathbb{Q}} = m_2$ exist. Define the set $O = \{u : m_1(u) - m_2(u) \geq 0\}$. Then we have

$$
\begin{aligned}
\mathbb{E}_{\mathbb{P}_1}[h(u_t)] - \mathbb{E}_{\mathbb{P}_2}[h(u_t)] &\leq \int h(m_1 - m_2)d\mathbb{Q} \\
&\leq \int_O h(m_1 - m_2)d\mathbb{Q} \leq \int_O M(m_1 - m_2)d\mathbb{Q} \\
&= M(\mathbb{P}_1(O) - \mathbb{P}_2(O)) \leq M\sup_O |\mathbb{P}_1(O) - \mathbb{P}_2(O)| \\
&= M\|\mathbb{P}_1 - \mathbb{P}_2\|_1 \leq M\sqrt{\frac{1}{2}D_{KL}\left(\mathbb{P}_2\|\mathbb{P}_1\right)}.
\end{aligned}
\tag{18}
$$

where the last inequality holds due to Pinsker's inequality. $\qquad\square$

# D DETAILS OF NEURAL ORACLES

We explain the details of the neural oracles discussed in Section 5. The results in this section are established in Deb et al. (2024), and we present them for completeness. First, we define the neural function class for which $\text{Alg}_R$ ensures a regret bound.

**Definition D.1** (Neural Function Class). *We consider the neural networks* $f_\theta : \mathbb{R}^d \mapsto \mathbb{R}$ *parameterized by* $\theta$:

$$f_\theta(x) = m^{-1/2}v^T\phi(m^{-1/2}W^{(L)}\phi(\cdots\phi(m^{-1/2}W^{(1)}x)))$$

*where* $W^{(1)} \in \mathbb{R}^{m \times d}$, $W^{(l)} \in \mathbb{R}^{m \times m}$ *for* $l \in \{2, \ldots, L\}$, $v \in \mathbb{R}^m$ *and* $\phi(\cdot)$ *is a Lipschitz and smooth activation function. We write* $\theta = (vec(W^{(1)})^T, \ldots, vec(W^{(L)})^T, v^T)^T \in \mathbb{R}^p$ ($p = md + (L-1)m^2 + m$) *and* $W^{(l)} = [w_{i,j}^{(l)}]$. *Based on this functional form, we define the function class* $\mathcal{F} = \{f_\theta : \theta \in B_{\rho,\rho_1}(\theta_0)\}$ *where* $B_{\rho,\rho_1}(\theta_0) = \{\theta : \|W^{(l)} - W_0^{(l)}\|_2 \leq \rho \text{ for } l \in [L] \; \|v - v_0\|_2 \leq \rho_1\}$ *for some initial parameter* $\theta_0 = (vec(W_0^{(1)})^T, \ldots, vec(W_0^{(L)})^T, v_0^T)^T$.

The neural function class $\mathcal{F}$ is a set of multi-layer perceptrons with depth $L$ and width $m$, whose parameters are $\ell_2$-norm bounded. This captures many widely used deep neural networks.

We need a specific initialization scheme to guarantee good properties of the neural function class $\mathcal{F}$.

**Assumption D.2** (Network Initialization). *The network parameters are initialized with* $w_{0,i,j}^{(l)} \sim \mathcal{N}(0, \sigma_0^2)$ *for all* $l \in [L]$ *where* $\sigma_0 = \frac{\sigma_1}{2(1+\sqrt{\log m/\sqrt{2m}})}$ *for some* $\sigma_1 > 0$, *and* $v_0$ *is a random unit vector.*

Additionally, a standard assumption on the positive definiteness of Neural Tangent Kernel (NTK) (Jacot et al., 2018) is required.

**Assumption D.3** (Positive Definite NTK). *The neural tangent kernel* $K_{NTK}(\theta) := \left[(\nabla f_\theta(x_i))^T\nabla f_\theta(x_j)\right]$ *is positive definite, i.e.* $K_{NTK}(\theta) \geq \lambda_0 I$ *for some* $\lambda_0 > 0$.

Our goal is to perform an online regression that guarantees Assumption 3.1. Formally, for each $t \in [T]$, we compute the estimator $\hat{f}_t$ to compete the best $f_\theta$ in hindsight:

$$\mathbb{E}\left[\sum_{i=1}^{t} \ell_i(\hat{f}_i) - \inf_{\theta \in B_{\rho, \rho_1}(\theta_0)} \sum_{i=1}^{t} \ell_i(f_\theta)\right] \leq \text{Regret}_R(t) \text{ for all } t \in [T].$$

Deb et al. (2024) shows that a projected Online Gradient Descent (OGD) in conjunction with random perturbation can serve as a regression oracle satisfying Assumption 3.1.

**Proposition D.4** (Theorem 3.2, 3.3 in Deb et al. (2024)). *Suppose that Assumption 5.1, D.2, and D.3 hold. If $\ell$ is square loss, there exists a projected OGD-based regression oracle that guarantees $\text{Regret}_R(t) \leq \mathcal{O}(\log T)$. If $\ell$ is logarithmic loss, by additionally assume that $y_t \in [z, 1-z] \, \forall t$ for some fixed $z > 0$, the oracle ensures $\text{Regret}_R(t) \leq \mathcal{O}(\log T)$.*

Using the regression oracle in Proposition D.4, DP-IGW achieves $\mathcal{O}(T^{\frac{2}{3}})$ regret upper bound.

# E DETAILS ON EXPERIMENTS

**Settings.** We explain details for $F_0$ and context distributions. Truncated Normal (TN) indicates $F_0 = TN(0, 0.2^2, -1, 1)$ where $TN(\mu, \sigma^2, a, b)$ is the truncated normal distribution of mean $\mu$, standard deviation of $\sigma$, and support $[a, b]$. Mixture of Uniform (MoU) indicates $F_0 = \frac{3}{4}U[-0.25, 0] + \frac{1}{4}U[0, 0.25]$. Since $p \in [0, 1]$ and $\beta^T x_t$ is zero-mean, we add bias to the linear model so that $\mathbb{P}(v_t > p \mid x_t) = 1 - F_0(p - (\beta^T x_t + 0.5))$. The PH model shows degenerate narrow distribution if the base CDF has narrow support, so we modify the truncated normal CDF to have wider support: $F_0 = TN(0, 1^2, -1, 1)$. In addition, we scale the true parameter for the PH model as $\mathbb{P}(v_t > p \mid x_t) = (1 - F_0(p))^{\exp(2\sqrt{d}\beta^T x_t)}$. The details on context distributions are as follows: normal distribution indicates $x_t \sim \mathcal{N}(0, \frac{1}{\sqrt{2d}}I)$, uniform in the unit ball indicates $x_t \sim U\{x : \|x\|_2 \leq 1\}$, and Bernoulli distribution indicates $x_{t,i} \sim \text{Ber}(0.5)$ for all $i \in [d]$.

**Neural Network Structure.** For DP-IGW, NeuralTS, NeuralUCB, SquareCB, and SmoothIGW, we use neural networks of the same structure. The networks consist of fully connected 3 layers, with input dimension $d + 1$ (context plus price) and output dimension 1, and hidden dimension $d + 1$. LeakyReLU activation with a negative slope 0.01 is used except in the output layer, where sigmoid activation is used. Adam (Kingma & Ba, 2014) optimizer with averaging coefficients $\beta_1 = 0.9, \beta_2 = 0.999$ is used with no weight decay. For each step, the optimizer performs 2 gradient steps with the loss computed with full-batch.

**Hyperparameter Search and Computational Resources.** Since every algorithm has hyperparameters to tune, we conduct a grid search on hyperparameters for $T_0 = 2000$ steps and report the result with the best hyperparameter and longer horizon $T = 5000$. For the experiments in Figure 3, longer steps of $T_0 = 3000, T = 30000$ are used.

For DP-IGW, we search with $\gamma \in \{4, 16, 64, 256, 1024\}$ and regression oracle learning rate $\alpha \in \{0.002, 0.01, 0.05\}$.

For Fan et al. (2022) and ExUCB (Luo et al., 2022), we optimize for $l_0 \in \{32, 64, 128, 256, 512\}$ and $C_1 \in \{\frac{1}{4}, \frac{1}{2}, 1, 2, 4\}$, where $l_0$ is the initial episode length and $C_1$ controls the ratio of exploration.

DEEP-C (Shah et al., 2019) also has two parameters $\gamma \in \{\frac{1}{4}, \frac{1}{8}, \frac{1}{16}, \frac{1}{32}, \frac{1}{64}\}$, where $\gamma$ is the confidence bound parameter.

For CoxCP (Choi et al., 2023), the search range is $l_0 \in \{64, 128, 256, 512, 1024\}$ where $l_0$ is the initial episode length.

For ABE (Chen & Gallego, 2021), we search over the exploration parameter $C \in \{\frac{1}{4}, \frac{1}{2}, 1, 2, 4\}$.

For SmoothIGW (Zhu & Mineiro, 2022) and SquareCB (Foster & Rakhlin, 2020), we search over exploration parameter $\gamma \in \{4, 16, 64, 256, 1024\}$ and $\gamma \in \{4, 16, 64, 256, 1024\}$, respectively.

For NeuralUCB (Zhou et al., 2020) and NeuralTS (Zhang et al., 2020), we search over $\gamma \in \{0.01, 0.1, 1, 10, 100\}$ and $\nu \in \{0.01, 0.1, 1, 10, 100\}$ where $\gamma$ is the confidence bound parame-

ter in NeuralUCB and $\nu$ is the sampling scale parameter in NeuralTS. Since SquareCB, NeuralUCB, and NeuralTS are finite-arm bandit algorithms, we discretize the price space with $K = 100$ arms for SquareCB, and $K = 20$ for NeuralUCB, NeuralTS (applying $K = 100$ requires too much computational resources).

For RMLP (Javanmard & Nazerzadeh, 2019), we optimize over $W \in \{1, 2, 4, 8, 16\}$, where $W$ is the $\theta$-norm constraint.

For ONSP (Xu & Wang, 2021), we search over $\gamma \in \{1, 4, 16, 64, 256\}$ and $\epsilon \in \{10^{-4}, 10^{-3}, 10^{-2}, 10^{-1}, 1\}$, where $\gamma$ is the Newton step update parameter and $\epsilon$ is the parameter for initial condition matrix.

The experiments were run on Intel Xeon Gold 6226R CPU and Nvidia GeForce RTX 3090 GPU, while our algorithm does not require high-throughput computational resources. Each run ($T = 5000$) completes within a few minutes.

### E.1 Real-world Datasets

We pre-processed all datasets by applying one-hot encoding for categorical features and normalized all numerical features to have zero mean and unit standard deviation. We also normalized the regression targets so that they have a mean of $0.5$ and a standard deviation of $0.25$.

**Abalone.** The Abalone Dataset (Nash & Ford, 1994) contains $4177$ data points to predict the age of abalone based on physical measurements. There are $8$ numerical features and one binary feature, which result in $d = 10$ dimensional contexts.

**Diamonds.** The Diamonds Dataset (Wickham, 2016) consists of $53940$ data points that measure the physical properties of diamonds. There are $7$ numerical features and $3$ categorical values, which lead to $d = 26$ dimensional contexts.

**Energy.** The Appliance Energy Prediction Dataset (Candanedo, 2017) has $19735$ data points for the prediction of energy consumption in a building based on numerical sensor measurements. There are $26$ numerical features in this dataset.

**Housing.** The California Housing Dataset (Pace & Barry, 1997) contains $20640$ data points to predict the median price of houses within a block. There are $9$ numerical features and one categorical feature, which result in $d = 13$ dimensional contexts. There are some missing values in numerical features, and we filled them with the mean value of each feature.

**Obesity.** The Estimation of Obesity Levels Dataset (Palechor & De la Hoz Manotas, 2019) aims to predict the obesity level of individuals based on their physical conditions and habits. There are $2111$ data points with $9$ numerical features and $d = 7$ categorical features, that forms $d = 23$ dimensional context vectors.

**Wine.** The Wine Quality Dataset (Cortez & Reis, 2009) consists of $4898$ data points with $d = 11$ numerical features. We used the data from white wine in the experiments.

## F Comparison of log-likelihood loss to square-loss

We compare the performance of `DP-IGW` with a log-likelihood oracle to one replaced with a square loss oracle. We experiment with the linear valuation model with normal/mixture of uniform CDFs and contexts sampled from normal/uniform distributions. We conduct a grid search for $T_0 = 2000$ steps and report the result with the best hyperparameter with a longer horizon $T = 5000$. Figure F shows the experimental results. The result demonstrates that `DP-IGW` with log-likelihood oracle consistently performs better than the one with square loss oracle, proving that the choice of log-likelihood loss is more suitable for regression on binary feedback.

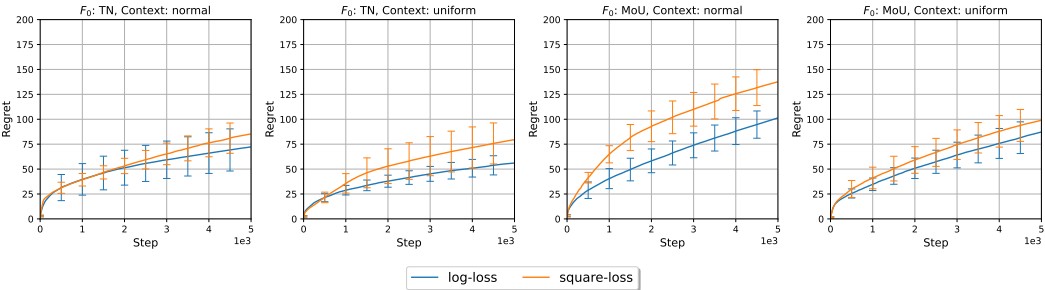

Figure 4: Cumulative regret (for $T = 5000$ steps) of `DP-IGW` with log-likelihood loss and square loss. For each algorithm, we executed 10 experiments and reported the mean and the standard deviation. Abbreviations each indicate TN: Truncated Normal, MoU: Mixture of Uniform.