# OpenReview forum: "Neural Dynamic Pricing: Provable and Practical Efficiency"
_ICLR.cc/2025/Conference — ICLR 2025 Conference Withdrawn Submission_

### Official Review · Reviewer_Dgya · 2024-11-01

**Soundness:** 3
**Presentation:** 3
**Contribution:** 3
**Rating:** 5
**Confidence:** 4

**Summary:**

The paper addresses limitations in current dynamic pricing (DP) algorithms, particularly those that rely on assumptions unfit for real-world applications and show poor data efficiency. The authors propose a new algorithm, DP-IGW, which incorporates neural networks as regression oracles and employs an exploration method called inverse gap weighting (IGW) to enhance data utilization and adaptability. Their method assumes minimal model assumptions, specifically Lipschitz continuity, and provides nearly optimal regret bounds, demonstrating both theoretical and practical efficiency across various settings.

**Strengths:**

1. By using neural networks as regression oracles, DP-IGW leverages their powerful generalization capabilities, allowing it to adapt to complex, high-dimensional data while maintaining provable guarantees.
2. The paper is technical sound. The authors provide a near-optimal regret bound, which establishes the efficiency and reliability of DP-IGW in theory.
3. The paper is well-written. The authors have a good summary of the current literature.

**Weaknesses:**

1. The design ideas and proof techniques seem to be very similar to Simchi-Levi and Xu (2022). Is there any technical challenge and contribution that the authors want to highlight?
2. For the lower bound (Section 5.2), I am wondering the purpose of the section. Usually, we are expecting a matched lower bound with the upper bound as Simchi-Levi and Xu (2022). The lower bound in Section 5.2 does not directly match with the upper bound, and even under different assumptions (i.e., the difference between Assumption 5.2 and Assumption 5.4). Does this mean that it is very hard to directly get a lower bound under Assumption 5.2?
2. Although the use of neural networks offers powerful generalization, it also introduces challenges like potential overfitting, sensitivity to hyperparameters, and higher computational demands. I will not push along this line, but want to point out this since the paper is a very practical relevant paper.

**Questions:**

Please see the weakness above.

---

> ### Author Response · Authors · 2024-11-29
>
> We appreciate the time and effort you have invested in reviewing our paper and providing thoughtful feedback. We hope our responses clarify your questions.
>
> 1. **Algorithmic Novelty:** Our DP-IGW exploits the dynamic pricing structure by separating the regression target (probability of purchase) and the exploration/optimization target (revenue). IGW-based bandit algorithms [1] use offline regression oracles that directly estimate reward (revenue), and then construct sampling distribution inversely proportional to the estimator. In contrast, our DP-IGW employs regression oracles that estimate the probability of purchase, and then construct sampling distribution inversely proportional to the estimated revenue. The idea of dividing estimating and exploration targets with the IGW technique was first proposed in our study, and we believe the idea can be applied to other sequential decision-making problems with complex regret structures (such as DP problems), for example, real-time bidding problems.
> 2. **Lower Bound:** The lower bound in Theorem 5.5 does match the upper bound in Theorem 5.3 up to logarithmic terms, as discussed in Line 287. The difference between Assumption 5.2 and Assumption 5.4 is not a concern for this minimax optimality, since Assumption 5.4 (for the lower bound) is stronger than Assumption 5.2 (for the upper bound). This means that the lower bound and proof of Theorem 5.5 are valid under Assumption 5.2. The additional constraint - Lipschitz continuity in context and monotonicity- in Assumption 5.4 reflects the properties of real-world dynamic pricing problems, thus Theorem 5.5 establishes the lower bound of dynamic pricing problems with Lipschitz continuity.
> 3. **Neural Network:** The challenges of neural networks have to be considered in practice. However, our experiments in Section 6 demonstrate consistent and stable performance of DP-IGW across an extensive range of environments. This is strong evidence of the fact that the performance of DP-IGW is not degraded by such potential concerns. The computational complexity for training neural networks is also reasonable, as our implementation of neural network oracle relies on standard logistic regression.
>
> [1] David Simchi-Levi and Yunzong Xu. Bypassing the monster: A faster and simpler optimal algorithm for contextual bandits under realizability. Mathematics of Operations Research, 47(3):1904–1931, 2022.

---

### Official Review · Reviewer_Lk57 · 2024-11-03

**Soundness:** 2
**Presentation:** 2
**Contribution:** 2
**Rating:** 3
**Confidence:** 3

**Summary:**

This paper designs a practical contextual DP algorithm that utilizes regression oracles. Our proposed algorithm assumes only Lipschitz continuity on the true conditional probability of purchase. It proves a $O(T^{2/3} reg_R(T)^{1/3})$ regret upper bound which is nearly minimax optimal in the canonical case of finite function class. Numerical experiments are conducted to verify the theory.

**Strengths:**

1. the presentation is clear.
2. the author provides both theoretical and experiments to verify their results.

**Weaknesses:**

The novelty is not enough.
 - Recent works have already considered using function approximation (e.g. [1]), so replacing such semi-parametric / non-parametric regression oracles with neural networks is not meaningful enough.
 - The authors state that their algorithm is fully sequential and flexible w.r.t. model assumptions, but both issues are tackled in the bandit settings, e.g., [2]. Since DP and bandits are extremely similar in problem structures, this paper is merely a combination of the two techniques and is not of much real significance.
 - The paper does not compare its assumption and regret with previous works, so no superiority of the proposed algorithm is illustrated.
 - The experiment does not highlight under which circumstances will it perform better than other models (e.g. log concavity of cdf $F_0$).


[1] Jianqing Fan, Yongyi Guo, and Mengxin Yu. Policy optimization using semiparametric models for dynamic pricing. Journal of the American Statistical Association, pp. 1–29, 2022.
[2] Yinglun Zhu and Paul Mineiro. Contextual bandits with smooth regret: Efficient learning in continuous action spaces. In International Conference on Machine Learning, pp. 27574–27590. PMLR,2022.

**Questions:**

No questions.

---

> ### Author Response · Authors · 2024-11-29
>
> We thank you for your time and effort in reviewing our paper. We carefully address your comments below, and hope that our responses clarify our key contributions.
>
> ### Significance of Neural Network Oracle
>
> We respectively disagree with the assessment that replacing semi-parametric / nonparametric oracle with neural networks is not meaningful. The neural network oracle enables robust and efficient learning across various environments, while providing provable regret bound with much weaker model assumptions compared to semi-parametric dynamic pricing methods. The power of neural network oracle is clearly presented in Section 6, where DP-IGW far outperforms existing semi-parametric / nonparametric baselines.
>
> - **DP-IGW with neural network is arguably the first DP algorithm that achieves both practical and statistical efficiency.**  As we demonstrated in Section 6, existing semi-parametric / nonparametric dynamic pricing (DP) methods fall short in terms of practical performance, despite having provable performance guarantees. DP-IGW with neural network oracle consistently outperforms baselines in every environments, including simulations and real-world datasets.
> - **The neural network oracle is more flexible than semi-parametric models**. Semi-parametric DP methods rely on the customer valuation model, where the probability of purchase is modeled by relatively restrictive function classes. For instance, the widely adopted linear valuation model assumes $P(y = 1 \mid x, p) = 1 - F_0 (p - \beta^T x)$. Moreover, some methods require additional assumptions on $F_0$ such as log-concavity or smoothness. In contrast, our neural network oracle allows a generic binary choice model $P(y=1 \mid x,p) = f(x,p)$ where $f$ is modeled by some neural network.
>
> ### Novelty of DP-IGW
>
> We would like to emphasize that the effective integration of bandit techniques into dynamic pricing (DP) problems is widely acknowledged in the DP literature for its significant contributions. The table below indicates the position of our work in the dynamic pricing literature in comparison to how previous works integrated bandit techniques into DP problems.
>
> | Bandit Literature | Dynamic Pricing Literature |
> | --- | --- |
> | UCB framework [3] | UCB-based noncontextual DP [12,17]
> UCB-based contextual DP [14,15] |
> | EXP4 [2] | EXP4-based DP [7,18] |
> | LASSO bandit [5] | DP using LASSO [10] |
> | Adaptive binning for bandits [16] | Nonparametric DP [6] |
> | Lower bound problem instances [9,13] | Lower bounds in [14,18] are based on [13], Lower bound in [11] is based on [9] |
> | Inverse Gap Weighting (IGW) [1,8,19] | **DP-IGW (ours)** |
>
> The contributions of these previous works are recognized by how the adaptation of bandit techniques to dynamic pricing captures the structure of DP. In this perspective, our work is based on the IGW technique [1,8,19] and DEC framework [8,19] for contextual bandits, but suffices substantial contributions as previous works have manifested. Our DP-IGW has the following distinct algorithmic features.
>
> - **DP-IGW exploits the dynamic pricing structure by separating the regression target (probability of purchase) and the exploration/optimization target (revenue).** IGW-based bandit algorithms [8,19] use regression oracles that directly estimate reward (revenue), and then construct sampling distribution inversely proportional to the estimator. In contrast, our DP-IGW employs regression oracles that estimate the probability of purchase, then construct sampling distribution inversely proportional to the estimated revenue. The idea of dividing estimating and exploration targets with the IGW technique was first proposed in our study.
> - **We design a practical anytime algorithm via epoch scheduling, which differs from existing oracle-based bandit algorithms.** Our method works without knowing the horizon $T$ a priori, hence it has a considerable advantage in practice. Our epoch scheduling differs from the standard doubling trick, which is a widely used technique for extension to an anytime algorithm. While the standard doubling trick requires resetting at every epoch to ensure independence between different epochs, our method does not reset itself every epoch but only adjusts the exploration parameter $\gamma_k$. This is possible because the decision-estimation coefficient (DEC) is bounded for every step, and this extension has not been tried in the related works.

---

> ### Author Response · Authors · 2024-11-29
>
> ### Regret Bound
>
> DP-IGW with neural network oracle achieves regret upper bound of $\tilde{O}(T^{2/3})$. This is sharper than the regret bounds of existing semi-parametric DP algorithms despite weaker assumptions and expressive model, as discussed in Section 5.1. We note that direct comparison of regret bound with previous semi-parametric DP methods is difficult, since the assumptions and models differ by method. Therefore, we emphasize that our upper bound is established with minimal assumptions: the realizability and Lipschitz continuity of the purchase model. The realizability assumption is common in DP / bandit literature, and the Lipschitz continuity assumption is much weaker compared to existing customer valuation models.
>
> ### Experiments with Baseline Methods
>
> The experiments in Section 6 consider an extensive set of environments, where some of them satisfy certain assumptions and some of them do not. In all environments, DP-IGW outperforms baselines regardless of whether their assumptions are satisfied or not.
>
> - We considered three semi-parametric valuation models in DP literature: linear, log-linear, and proportional hazard models. The evaluation of each baseline method is conducted with the model each method assumes, thus there is no model misspecification for baseline methods.
> - Truncated Normal CDF ($F_0$) satisfies both smoothness and log concavity conditions while Mixture of Uniform CDF does not.
> - Real-world datasets have complex structures that do not fall into certain purchase models. Therefore, the results on the real-world datasets indicate generalization capability to model misspecification.
>
> [1] Abe, N., & Long, P. M. (1999, June). Associative reinforcement learning using linear probabilistic concepts. In *ICML* (pp. 3-11).
>
> [2] Auer, P., Cesa-Bianchi, N., Freund, Y., & Schapire, R. E. (2002). The nonstochastic multiarmed bandit problem. *SIAM journal on computing*, *32*(1), 48-77.
>
> [3] Auer, P., Cesa-Bianchi, N., & Fischer, P. (2002). Finite-time analysis of the multiarmed bandit problem. *Machine learning*, *47*, 235-256.
>
> [5] Bastani, Hamsa, and Mohsen Bayati. "Online decision making with high-dimensional covariates." *Operations Research* 68.1 (2020): 276-294.
>
> [6] Chen, N., & Gallego, G. (2021). Nonparametric pricing analytics with customer covariates. *Operations Research*, *69*(3), 974-984.
>
> [7] Cohen, M. C., Lobel, I., & Paes Leme, R. (2020). Feature-based dynamic pricing. *Management Science*, *66*(11), 4921-4943.
>
> [8] Foster, D., & Rakhlin, A. (2020, November). Beyond ucb: Optimal and efficient contextual bandits with regression oracles. In *International Conference on Machine Learning* (pp. 3199-3210). PMLR.
>
> [9] Goldenshluger, A., & Zeevi, A. (2009). Woodroofe’s one-armed bandit problem revisited.
>
> [10] Javanmard, A., & Nazerzadeh, H. (2019). Dynamic pricing in high-dimensions. *Journal of Machine Learning Research*, *20*(9), 1-49.
>
> [11] Keskin, N. B., & Zeevi, A. (2014). Dynamic pricing with an unknown demand model: Asymptotically optimal semi-myopic policies. *Operations research*, *62*(5), 1142-1167.
>
> [12] Kleinberg, R., & Leighton, T. (2003, October). The value of knowing a demand curve: Bounds on regret for online posted-price auctions. In *44th Annual IEEE Symposium on Foundations of Computer Science, 2003. Proceedings.* (pp. 594-605). IEEE.
>
> [13] Kleinberg, R. (2004). Nearly tight bounds for the continuum-armed bandit problem. *Advances in Neural Information Processing Systems*, *17*.
>
> [14] Luo, Y., Sun, W. W., & Liu, Y. (2022). Contextual dynamic pricing with unknown noise: Explore-then-ucb strategy and improved regrets. *Advances in Neural Information Processing Systems*, *35*, 37445-37457.
>
> [15] Luo, Y., Sun, W. W., & Liu, Y. (2024). Distribution-free contextual dynamic pricing. *Mathematics of Operations Research*, *49*(1), 599-618.
>
> [16] Perchet, V., & Rigollet, P. (2013). The multi-armed bandit problem with covariates. *The Annals of Statistics*.
>
> [17] Wang, Y., Chen, B., & Simchi-Levi, D. (2021). Multimodal dynamic pricing. *Management Science*, *67*(10), 6136-6152.
>
> [18] Xu, J., & Wang, Y. X. (2022, May). Towards agnostic feature-based dynamic pricing: Linear policies vs linear valuation with unknown noise. In *International Conference on Artificial Intelligence and Statistics* (pp. 9643-9662). PMLR.
>
> [19] Zhu, Y., & Mineiro, P. (2022, June). Contextual bandits with smooth regret: Efficient learning in continuous action spaces. In *International Conference on Machine Learning* (pp. 27574-27590). PMLR.

---

### Official Review · Reviewer_1mmY · 2024-11-05

**Soundness:** 2
**Presentation:** 3
**Contribution:** 2
**Rating:** 5
**Confidence:** 3

**Summary:**

The authors introduce a contextual dynamic pricing algorithm called DP-IGW, which aims to address some of the limitations in the existing dynamic pricing literature, such as strong modeling assumptions or poor performance under real-world conditions.
The idea of DP-IGW is to leverage regression oracles (in particular, neural networks) to enable contextual dynamic pricing.
The main assumption is that the function $f^\star$ that maps the contexts $x$ and prices $p$ to the probability of selling at a price $p$ when the context is $x$ is uniformly Lipschitz in the prices (Assumption 5.2 --- note that it is assumed that the same $L$ works for all contexts).
The authors prove that the algorithm achieves an upper regret bound of order $T^{2/3} \mathrm{Regret}_R(T)^{1/3}$, where $\mathrm{Regret}_R(T)$ is the regret after $T$ time steps of the regression oracle used by DP-IGW.
The regret rate is proved to be optimal only in the special case where $f^\star$ belongs to a known _finite_ family and the regression oracle only outputs regressors in this family.
Experiments illustrate the theoretical findings.

**Strengths:**

- The paper is mostly clear in its descriptions, presenting both algorithmic and theoretical insights with sufficient effectiveness.

- The problem is of broad interest, and the idea of mixing neural network technology with dynamic pricing is appealing.

**Weaknesses:**

The originality of the contribution appears somewhat limited, given that the key ideas are adaptations of somewhat standard techniques. This is not necessarily a reason to reject a paper. In fact, it could even be a plus to have a simple change that leads to a large improvement. It is not clear to me that this is the case here, though. The lower bound near-matches the upper bound only in a very narrow case (finite $\mathcal F$), and I am not sure about the optimality of this approach in the general case.

**Questions:**

- I suggest the authors close the gap between the upper and lower bound in the general case, or at least in a significantly broader case.

- Can the assumption be relaxed to: for all contexts $x$, there exists a Lipschitz constant $L_x$ such that $f^\star(x,\cdot)$ is $L_x$-Lipschitz? What about piece-wise Lipschitz?

---

> ### Author Response · Authors · 2024-11-29
>
> Thank you for dedicating your time and expertise to reviewing our work and for offering constructive feedback. Below, we provide detailed responses to address your comments and questions.
>
> ### Novelty and Contribution
>
> - Our algorithm is technically novel compared to previous works in two main perspectives. First, our algorithm exploits the dynamic pricing structure by separating the regression target (probability of purchase) and the exploration/optimization target (revenue), differentiating it from IGW-based bandit algorithms. The idea of dividing estimating and exploration targets with the IGW technique was first proposed in our study, and we believe the idea can be applied to other sequential decision-making problems with complex regret structures.  Second, our algorithm is a practical any-time algorithm via epoch scheduling, which differs from the standard doubling trick. Compared to the doubling trick, epoch scheduling does not reset itself every epoch but only adjusts the exploration parameter $\gamma_k$. This is possible because the decision-estimation coefficient (DEC) is bounded for every step, and this extension has not been tried in the related works.
> - Our algorithm first proposed the integration of regression oracles, especially the neural regression oracle, into the DP problem, which led to a significant performance enhancement in extensively wide environments. Our work shows provable guarantees, and the extensive experiment results support that our algorithm is the most practical contextual DP algorithm which performs well across various instances.
>
> ### Lower Bound
>
> We appreciate the reviewer’s thoughtful feedback and the opportunity to clarify the originality and significance of our contributions. While it is true that our lower bound result in Theorem 5.5 does not directly provide a matching bound for more general function classes, we would like to emphasize that this has no bearing on the originality of our work. The originality of our contribution lies in the development of a practical and theoretically sound contextual DP algorithm that achieves strong performance across various function classes, rather than solely in achieving matching lower bounds.
>
> Our work brings novelty to the literature in several key aspects:
>
> - Our regret upper bound for general function classes extends the understanding of contextual DP algorithms in settings where prior works have been either overly restrictive or lacked practical applicability. This contribution fills an important gap in the literature.
> - Practical Effectiveness: Through extensive experiments, we demonstrate that our method performs consistently well across a wide range of circumstances, making it a robust and practical solution. Importantly, the design and implementation of our algorithm are informed by its theoretical foundations, ensuring a balance between rigor and real-world usability.
>
> Matching lower bounds are not the primary focus of our work, and this does not detract from the originality or significance of our contributions.
>
> We sincerely hope this clarification highlights our perspective and demonstrates the originality and value of our contributions to the field.
>
> ### Lipschitz Continuity
>
> - The Lipschitz continuity can be relaxed to the assumption you suggested. However, the corresponding regret bound would include $\sup_x L_x$ instead of $L$.
> - The Lipschitz continuity is required to prove Lemma A.2 (relation between smooth regret and standard regret). If we assume the context-dependent Lipschitz constant $L_x$, we have to replace the constant $L$ in equation 4 with $\sup_x L_x$ , as the sequence $ (x_t )_{t≤T}$ is arbitrary.
> - Since context-dependent Lipschitz continuity implies uniform Lipschitz continuity with the constant $L = \sup_x L_x$, this relaxation of assumption does not improve the regret bound.

---

> > ### Comment · Reviewer_1mmY · 2024-12-02
> >
> > Thank you for your response. Unfortunately, I was not sufficiently swayed, and I will maintain my score.

---

### Note · Authors · 2025-01-21

I have read and agree with the venue's withdrawal policy on behalf of myself and my co-authors.